# Noninvasive imaging-guided ultrasonic neurostimulation with arbitrary 2D patterns and its application for high-quality vision restoration

Gengxi Lu [1,2,4], Chen Gong [1,2,4], Yizhe Sun [1], Xuejun Qian [1,2], Deepthi S. Rajendran Nair [2], Runze Li [1,2], Yushun Zeng [1], Jie Ji[1,2], Junhang Zhang [1,2], Haochen Kang [1], Laiming Jiang[1], Jiawen Chen[1], Chi-Feng Chang[1,2], Biju B. Thomas [2], Mark S. Humayun[1,2,3] & Qifa Zhou [1,2,3] ✉

Retinal degeneration, a leading cause of irreversible low vision and blindness globally, can be partially addressed by retina prostheses which stimulate remaining neurons in the retina. However, existing electrode-based treatments are invasive, posing substantial risks to patients and healthcare providers. Here, we introduce a completely noninvasive ultrasonic retina prosthesis, featuring a customized ultrasound two-dimensional array which allows for simultaneous imaging and stimulation. With synchronous three-dimensional imaging guidance and auto-alignment technology, ultrasonic retina prosthesis can generate programmed ultrasound waves to dynamically and precisely form arbitrary wave patterns on the retina. Neuron responses in the brain's visual center mirrored these patterns, evidencing successful artificial vision creation, which was further corroborated in behavior experiments. Quantitative analysis of the spatial-temporal resolution and field of view demonstrated advanced performance of ultrasonic retina prosthesis and elucidated the biophysical mechanism of retinal stimulation. As a noninvasive blindness prosthesis, ultrasonic retina prosthesis could lead to a more effective, widely acceptable treatment for blind patients. Its real-time imaging-guided stimulation strategy with a single ultrasound array, could also benefit ultrasound neurostimulation in other diseases.

Visual impairment and blindness, impacting over 200 million individuals globally, pose a significant public health challenge[1]. A principal cause of vision loss is photoreceptor degeneration (PD) induced by ocular diseases such as age-related macular degeneration, retinitis pigmentosa, and diabetic retinopathy. Most of these conditions are incurable and irreversible, leading to lifetime vision impairment. The only existing approach to treat them involves using visual prostheses, which provide artificial vision by electrically stimulating the remaining neurons in the visual pathway. Visual prostheses can be categorized based on their position within the visual pathway: the retina[2–6], optic

[1]Department of Biomedical Engineering, University of Southern California, Los Angeles, CA, USA. [2]Roski Eye Institute, Keck School of Medicine, University of Southern California, Los Angeles, CA, USA. [3]USC Ginsburg Institute for Biomedical Therapeutics, University of Southern California, Los Angeles, CA, USA. [4]These authors contributed equally: Gengxi Lu, Chen Gong. ✉e-mail: qifazhou@usc.edu

nerve[7], or cortex[8]. Retinal prostheses are typically preferred due to their simple visual mapping and the ability to use the natural signal processing of the visual pathway.

Although significant progress has been made in recent decades to advance the resolution and other performance aspects of retinal prostheses[2–6], existing approaches are notably invasive, requiring electrode implantation and surgical intervention, thereby posing significant risks and burdens to patients. Moreover, the limited field of view (FOV) of electrode-based prostheses, which is less than 30°, prevents patients from efficiently performing daily activities[9]. Other less-invasive techniques, such as optogenetics[10] and sonogenetics[11], take advantage of genetic engineering to provide more precise and specific modulation of neurons. Despite their advancements, these techniques require complicated genetic engineering pre-treatments and the introduction of viruses into the body. As these techniques are still clinically inconclusive, and potential side effects are not fully understood, their clinical applications remain limited.

Ultrasound neurostimulation, pioneered in 1958[12], is an emerging completely noninvasive technology that requires no pretreatment. With a resolution at the micrometer level and a penetration depth of centimeters in the human body, ultrasound neurostimulation holds significant promise in treating various diseases[13–15]. Recent studies showed that ultrasound stimulation on the retina can activate neurons between photoreceptors and retina ganglion cells (RGCs), suggesting the feasibility and potential of a noninvasive ultrasound-based retina prosthesis[16–18]. However, the progress in translating ultrasound stimulation from laboratory experiments to a clinical treatment for everyday use is slow and challenging. Since neurostimulation requires a long-term treatment, imaging guidance is necessary to ensure the precise ultrasound stimulation on the targeted neurons. Existing methods use magnetic resonance imaging[19] or ultrasound imaging using another imaging probe[14] to locate the stimulation target and position the array correctly. Both methods require frequent and manual calibrations and are impractical for daily use as personal healthcare devices. Also, to cover the targeted neuron during inevitable body movements, the stimulation focal size is relatively large[13–15], leading to higher risks of side effects. Furthermore, visual restoration specifically requires generating arbitrary and comprehensive stimulation patterns, creating dynamic visual patterns with minimal latency.

To address these issues and demonstrate the translational potential of ultrasound retina stimulation from research to clinics, we introduced the ultrasound-based retina prosthesis (U-RP, Fig. 1a–c) and investigated its feasibility and performance in blindness treatment. The concept of U-RP starts with capturing external environmental images through a camera. These images are then relayed to a processing and control unit where they are converted into ultrasound control signals, directing the ultrasound array. Subsequently, the specialized two-dimensional (2D) ultrasound array generates ultrasound waves and forms patterns to stimulate the retina. The stimulated retinal neurons generate visual signals, which are conveyed to the brain via the optic nerves, resulting in artificial visions. To precisely stimulate the retina in vivo, we facilitated 3D-imaging-guided dynamic-pattern stimulation on the retina, effectively recreating corresponding neuronal patterns over a wide field of view (FOV). Additionally, we thoroughly examined the frequency-dependent performance, safety parameters, and underlying physical mechanisms of ultrasound retinal stimulation.

## Results
### Design and validation of U-RP
The essential component of the U-RP is the ultrasound 2D array. Using a single ultrasound probe for both imaging and stimulation is usually challenging due to the contrasting requirements for parameter optimization. Active elements with a high mechanical quality factor and low loss are optimal for stimulation which requires substantial power,

whereas imaging elements prefer a low mechanical quality factor to achieve superior resolution. Nevertheless, we designed and fabricated a 16-by-16 channel 4.5-MHz ultrasound array, striking an optimized balance between imaging and stimulation (Supplementary Fig. 2, Supplementary Table. S1). The fabricated array was characterized (Supplementary Figs. 3 and 4) and demonstrated the capability of imaging eyeball structures, as well as generating sufficient power to stimulate retina neurons in vivo.

We initially verified that ultrasound stimulation on the retina could generate artificial visual signals in rats (Fig. 1d). To evaluate whether ultrasound-evoked retinal activity propagates to higher visual centers, we used a customized multi-electrode array (MEA) to map neuron activities on the top surface of the contralateral superior colliculus (SC), a visual center in the midbrain that receives feedforward connections from optic nerve[20]. We selected the SC over the visual cortex due to its straightforward topological relationship with the retina, which facilitated a delicate comparison between ultrasound stimulation patterns and ultrasound-induced visual patterns. In addition to electrophysiological recording on SC, fiber photometry was also performed to monitor neuron responses at the primary visual cortex (Supplementary Fig. 5a) to further corroborate the existence of artificial vision.

Both healthy and visually impaired rats were used in this experiment, differentiated by the histological difference in their retinas. The absence of outer nuclear layers in the PD retina indicated the loss of photoreceptors. The experiment began with administering a full-field light stimulus to verify the functional vision of the rats. Healthy rats exhibited stable neuron response at the SC, while blind rats displayed no neuron responses (Fig. 1e). Upon switching to focused ultrasound stimuli (4.5 MHz, 10 ms, 2.83 MPa), both healthy and blind rats showed repetitive neuronal responses (Fig. 1e, Supplementary Fig. 5b, c). The amplitude of ultrasound-induced responses was similar to those induced by light (Fig. 1f).

Previous studies raised the concern that ultrasound stimulation might inadvertently activate non-targeted brain regions by inducing skull vibrations and causing widespread artifacts[21,22]. To address it, we conducted two groups of control experiments. In the first group, the ultrasound probe was tilted to focus between the eye and ear of the rat. In the second group, MEA was inserted 1.5 mm deeper than the surface of SC. No ultrasound-evoked neuron activity was observed in both control groups. This confirmed that ultrasound precisely targeted the retina and only evoked vision-related neuron responses (Supplementary Fig. 6 and Supplementary Fig. 5d).

### Auto-aligned pattern stimulation via ultrasound 3D imaging guidance
Achieving precise ultrasound stimulation is desired yet challenging in clinical scenarios. The difficulty arises from factors such as imprecision of array positioning, variations of organ size, and tissue movements (Fig. 2a, Supplementary Fig. 7). To ensure accurate and effective stimulation on the retina, we implemented an auto-alignment technique in U-RP, relying on ultrasound 3D imaging and automated position detection. Using the same ultrasound 2D array, cross-sectional imaging of eyeball in XZ and YZ planes were captured (Fig. 2b–d). Although the imaging quality of this array was not as high as that of arrays specialized for imaging, the cornea and retina shapes were distinctly visualized. This sufficient image contrast was owing to the balanced imaging ability of the ultrasound array and ultra-low reflection of the vitreous body in the eyeball. Following image acquisition, automatic edge detection was performed to precisely ascertain the location of the retina. A 3D model of the eyeball and its boundaries were reconstructed by aggregating layered 2D images. The distance and tilt angle (Fig. 2e, f) from the array surface to the retina were extracted and subsequently incorporated into the pattern generation algorithm as space correction feedback (Fig. 3a).

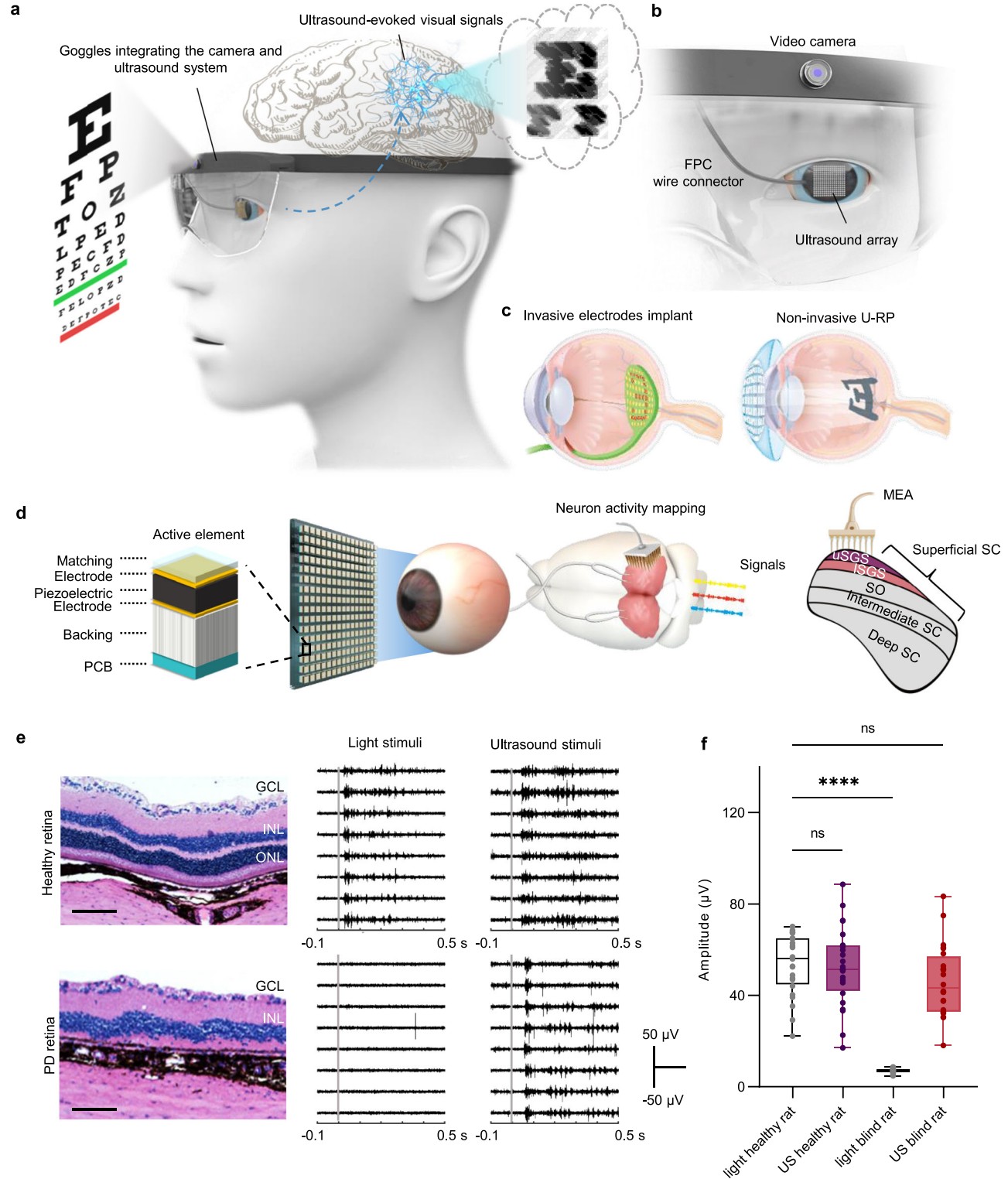

Ultrasound 2D array can dynamically generate arbitrary patterns at desired locations by electrically controlling the amplitude and phase of each channel (Supplementary Fig. 8). The close-loop workflow of precise pattern stimulation is outlined (Fig. 3a). The amplitude and phase distributions were computed by solving the Rayleigh-Sommerfeld equation (see Methods). The retrieved depth and tile angle from imaging were also considered to ensure the precise formation of stimulation patterns on the retina. Subsequently, local retinal neurons were activated, and neuron responses with the same pattern were observed on SC (Fig. 3b). For further illustration, we

showcased the replication of letters "U", "S", "C" and symbols for "Turn right", "Stop" (Fig. 3c–f). Raw images were decoded into amplitude and phase distributions to drive the 2D array (Fig. 3d). The resultant ultrasound fields in the free field were then simulated (Supplementary Fig. 9) and mapped by hydrophone system (Fig. 3e), yielding consistent patterns. Subsequently, ultrasound was used to stimulate the retina and evoke the retinal neurons with the same patterns (Fig. 3f, Supplementary Fig. 10). To gauge the quality of ultrasound pattern stimulation, we calculated the correspondence between ultrasound stimulation patterns on the retina and the neuron responses mapped

**Fig. 1 | U-RP noninvasively activating PD retinas. a** The conceived schematic of U-RP: The ultrasound array is worn as a contact lens to noninvasively transmit ultrasound waves into the eyeball. Activated retinal neurons generate visual signals and the brain processes signals into meaningful visual patterns. **b** The front view of the conceived U-RP. **c** Existing retina prostheses require surgery and invasive electrode implants, while U-RP is attached to the surface of the eyeball and non-invasively stimulates the whole retina. **d** The schematic of in vivo experiment setup on rats: the ultrasound probe (or the light flasher) is attached to the surface of the eyeball to stimulate the retina, and the MEA records neuron responses from the surface of SC. (PCB: printed circuit board, uSGS: upper superficial gray layer, lSGS: lower superficial gray layer). **e** The histology image of the retina from a healthy (left top) and a blind rat (left bot). (GCL: Ganglion cell layer; INL: Inner nuclear layer; ONL: outer nuclear layer.) The scale bar is 0.1 mm. The repetitive neuron responses recorded on the top surface of SC from the healthy rat (mid) and the blind rat (right). Ultrasound parameters for retina stimulation: 4.4 MHz, 10 ms, 3 MPa. Experiments were repeated independently on three healthy and three blind rats. Each line shows signals from one repetition. Gray solid lines indicate the beginning of ultrasound stimulation. **f** Statistical analysis of amplitudes of neuron responses. The box extends from the 25th to 75th percentiles and the line in the middle of the box is plotted at the median. Whiskers cover from the minimum to maximum and all data points. $n = 24$ independent experiments from three animals. $p = 0.9828$(ns), <0.0001 (****), and 0.1516(ns). Ordinary one-way ANOVA and Dunnett's multiple comparison tests were performed to compare the results versus light healthy rat group. Source data are provided as a Source Data file.

on the SC surface. The average similarity between neuronal response mapping and input images (see Methods) was $0.91 \pm 0.03$, ($n = 8$, s.e.m.).

While some might concern that internal structures of the eyeball would impede ultrasound propagation and distort the reconstructed patterns on the retina, our results indicated that these concerns were largely unnecessary. Acoustic refraction and reflection, primarily induced by variations in sound speed and acoustic impedance, respectively, could indeed cause field distortion. However, due to the small variations in acoustic properties within the eyeball (Supplementary Table. S2), the impact of these distortions was negligible. A quantitative evaluation of this distortion is addressed in a subsequent section on spatial resolution.

### Performance characterization of U-RP

Just as the electrode size determines the spatial resolution in electrical retinal prostheses, the lateral size of the ultrasound focal point on the retina determines the spatial resolution of U-RP, a crucial parameter for visual prostheses. We quantified the spatial resolution of U-RP as the full width at half maximum pressure (FWHM) of ultrasound focus. Since the focal depth is limited by the geometric size of the eyeball, FWHM is determined by the aperture size and frequency (Fig. 4a). In this experiment, four ultrasound transducers with center frequencies of 3, 5.4, 12, and 20 MHz were used to demonstrate high-resolution retina stimulation (Supplementary Figs. 11 and 12). With an increase in center frequency, the FWHM measured in free field decreased from 590 μm to 81 μm, resulting in an estimated visual acuity of 20/460 at 12 MHz and 20/320 at 20 MHz (Fig. 4b). Finite element method (FEM) simulations were conducted in both free field and eyeball (Fig. 4b and Supplementary Figs. 1, 13 and 14) to estimate the distortion in ultrasound field from eyeball structures. The distortion quantified the change in FWHM, which was lower than 4%.

FOV is another important consideration for retina prostheses. However, it is challenging to increase the FOV of electrode-based prostheses due to the increased electrode number and system complexity. As a non-invasive modality with a steerable focus, U-RP inherently has a full-size FOV. We performed both simulations and experiments, demonstrating that U-RP could stimulate at least a 6-mm-by-6-mm area, which covered the whole retina of rats (Fig. 4c–e, Supplementary Fig. 14). Neuron response mapping also showed that the visual responses scanned through the whole reception field on SC (Fig. 4e).

The temporal resolution of ultrasound retina stimulation was determined by the highest stimulation rate that can consistently evoke stable neuron responses. This temporal resolution defines the available frame rate of U-RP, another crucial parameter for visual prostheses. The previous study using 3.1 MHz ultrasound[23] observed neuron fatigue at 10 Hz. In this experiment, we tested temporal resolution with various ultrasound frequencies (Fig. 4f, g). The available frame rate of U-RP increased from 5 Hz to 15 Hz as the stimulation area decreased. Despite a high repetition rate, no damage was observed, as the neuron fatigue was reversible after several seconds of rest.

### Physical mechanism of U-RP

The understanding of the biophysical mechanism of ultrasound retina stimulation could play a key role in guiding future studies. Physical mechanisms can be revealed by investigating frequency-dependent efficiency. The prior study on ex vivo retina demonstrated that higher-frequency ultrasound was more efficient to evoke retinal neuron responses[16]. However, this has not yet been validated in vivo. We revisited this question by measuring the frequency-dependent efficiency of ultrasound retina stimulation, identified as the threshold of derated acoustic pressures on the retina for stable retinal neuron activation. By driving ultrasound transducers within their frequency bandwidth (Supplementary Fig. 11), the frequency-dependent curve of threshold was obtained (Supplementary Fig. 15). The measured pressure threshold was inversely proportional to the frequency, with a root mean squared error (RSME) of 0.035. This indicated acoustic cavitation was not dominant because cavitation effects are more efficient when frequencies are lower. To distinguish the role of ultrasound-induced thermal effect and acoustic radiation force (ARF), we simulated and measured the temperature increase in ex-vivo eyeballs. Only a minor temperature increase of less than 0.5 °C (Supplementary Fig. 15b) was observed, which is insufficient to affect neuronal activities. Therefore, our results provided in vivo evidence to support that ARF is the physical mechanism for retina stimulation.

### Guidance of rat behavior using ultrasound stimuli as visual signals

After substantiating the efficacy of U-RP with electrophysiological characterizations, we further conducted behavioral tests to ascertain the functional efficacy of ultrasound retina stimulation. In the water-licking tests, either pointed light or focused ultrasound stimuli acted as cues for delayed water drops for water-restricted and head-fixed animals (Fig. 5a, Supplementary Fig. 16). The water-licking behavior was recorded and counted (Fig. 5b&c). In the first set of tests, PD rats were trained with ultrasound stimuli, and their anticipatory lick rate (ALR) was evaluated, defined as the lick rate after the stimulus and before the water appearance. Over seven days, these PD rats progressively learned to associate the ultrasound stimuli with water drops, as indicated by the ALR increase curve (Fig. 5d). However, the concern was raised that the ultrasound might also trigger mechanical force or thermal effect on the retina and that these sensations, not visual signals, could be what the PD rats associated with the water drops.

To address this concern, we conducted a second set of tests on healthy rats that had been trained to anticipate water drops after light stimulus (Fig. 5c, right). On Day 8, we switched the light stimulus to an ultrasound stimulus. No significant difference ($n = 8$, s.e.m.) in ALR was observed (Fig. 5d), indicating that ultrasound stimulation on retinas effectively generated visual signals similar to those from light stimuli. To rule out any unintended cues like device noise or random mechanical stimulus, we conducted control tests, in which the ultrasound stimulated the skin between the eyeball and ear. ALR did not increase in these controls (Fig. 5e).

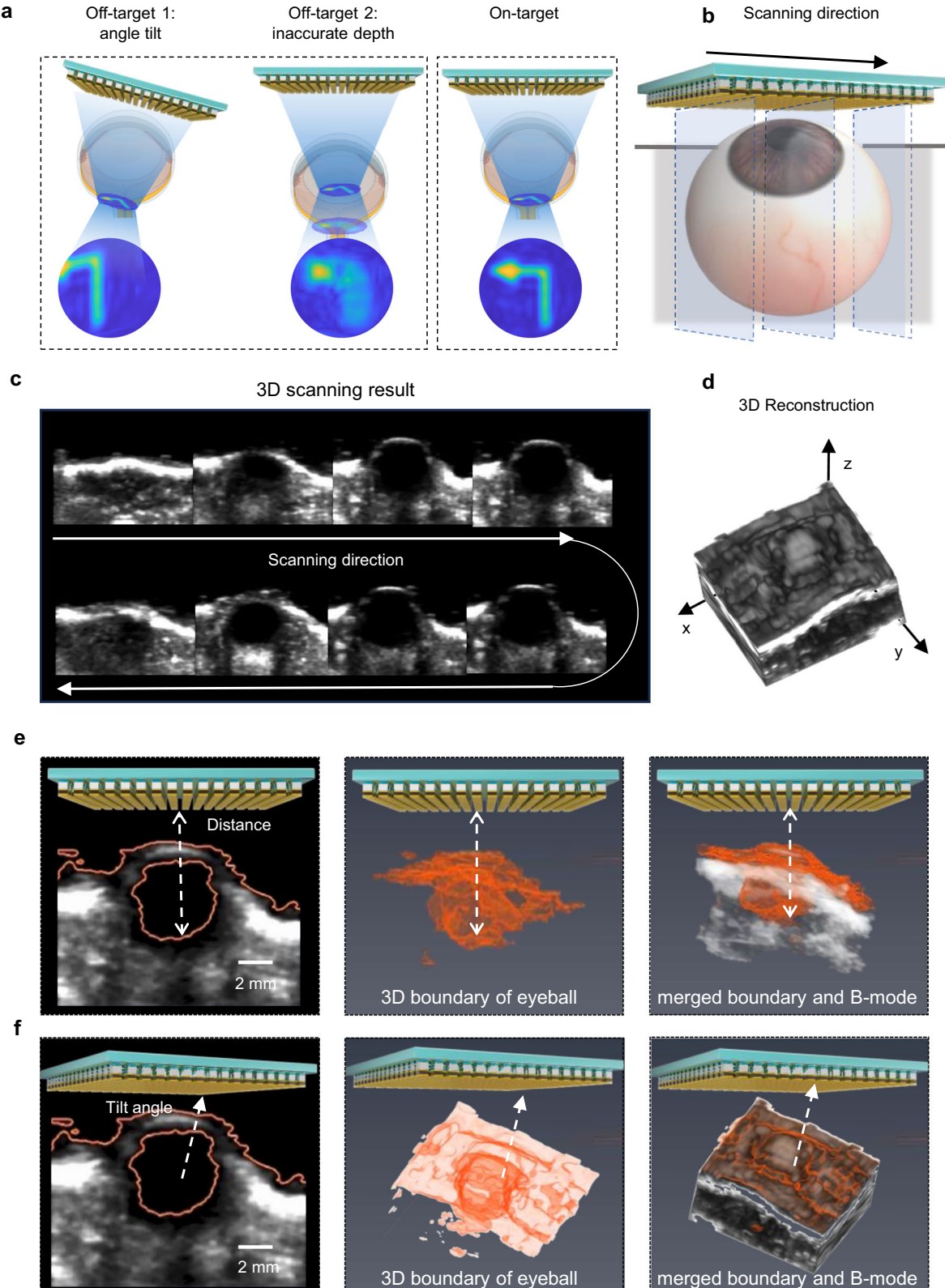

**Fig. 2 | Precise stimulation of retina using 3D imaging guidance and retina auto-detection. a** Illustrations showing that the ultrasound stimulation pattern on the retina would be significantly compromised in off-target situations. When angle is tilt, the stimulation pattern would be tilted and off-axis. When focal depth is inaccurate, the stimulation pattern would be blurred. **b** Schematic showing ultrasound 2D array electrically scanning the eyeball to reconstruct its 3D model. **c** Representative examples of acquired B-mode images in the scanning direction. **d** Reconstructed 3D model of the rat eyeball. **e, f** Cornea surface and retina highlighted by the retina auto-detection algorithm, and the calculated relative distance **e** and angle **f** between the array surface and retina. Experiments were repeated independently two times on one blind rat and one healthy rat.

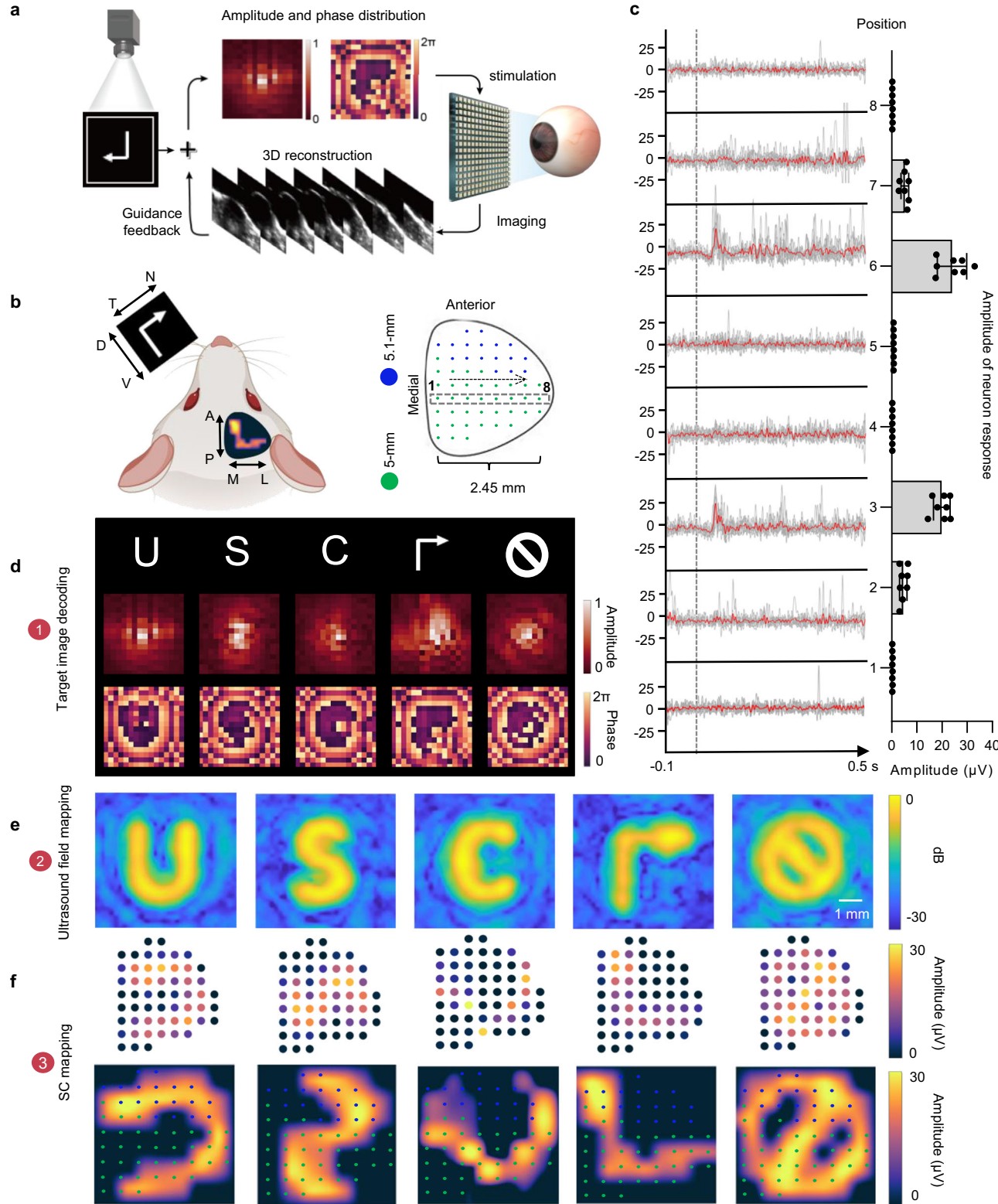

## Safety examination of U-RP

To evaluate the safety aspects of U-RP, we conducted comprehensive examinations following long-period ultrasound stimuli. In this experiment, ultrasound stimulation was administered at three frequencies (3, 4.5, and 20 MHz) and pressure amplitudes 50% higher than the threshold were conducted for 3 hours every three days over two weeks on healthy rats. Firstly, to ascertain whether there was any structural damage or blurring in the entire eyeball as a result of ultrasound stimuli, we conducted optical coherence tomography (OCT) and fundus imaging before and after the stimulation. Clear imaging of the retina suggested the eyeball remained unobstructed and did not show any structural damage to the retina (Supplementary Fig. 17a, b). Secondly, we measured electroretinography (ERG) responses to full-field light stimuli to evaluate any potential functional damage in photoreceptors and bipolar cells caused by ultrasound stimulation (Supplementary Fig. 17c). Neither the amplitudes nor time delays of a- and

**Fig. 3 | Visual pattern regeneration using U-RP. a** The feedback system of precise ultrasound pattern generation: Ultrasound 2D array acquires 3D images of the eyeball, detects retina location. Arbitrary target image was decoded to ultrasound amplitude and phase distributions. Retina location (distance and angle) was considered by the decoding algorithm to modify distributions. The modified ultrasound amplitude and phase distributions were used to drive ultrasound 2D array. **b** An illustration of the retinotopic projection to the SC, and the spatial relation between the customized MEA and the surface of SC. "N", "T", "D", and "V" stand for "Nasal", "Temporal", "Dorsal" and "Ventral". "A", "P", "L", and "M" stand for "Anterior", "Posterior", "Lateral" and "Medial". **c** Representative signals mapped by the fourth-column electrodes of MEA, when stimulated by the 'C' pattern. Gray solid lines indicate the raw signal is each recording and red lines indicate the averaged signal. Gray dotted lines indicate the beginning of ultrasound stimulation ($n = 8$ independent experiments). The histogram shows the amplitude distribution. Median, Error bar: s.d., $n = 8$ independent experiments. **d–f** Generation of five different visual patterns "u", "s", "c", "turn right", and "stop": Target images and decoded ultrasound amplitude and phase distributions **d**, free-field ultrasound patterns mapped by hydrophone **e**, mapping of neuron responses on the surface of SC **f**. Top row shows responses from each electrode of MEA. Bottom row shows responses after 2D interpolation.

b-waves showed any statistically significant changes (Supplementary Fig. 17d, n = 8, s.d.), demonstrating intact function of photoreceptors and bipolar cells. Since repetitive and stable neuron responses were observed from the brain during and after ultrasound stimulation, the function of RGCs was inferred to be intact as well. At the end of the experiment, stimulated eyeballs were collected and evaluated with Hematoxylin & Eosin (H&E) staining and immunostaining. Ultrasound stimulation neither changed fundus structures, reduced retinal layer thickness, activated microglia, increased apoptosis, nor compromised the function of photoreceptors. (Supplementary Fig. 17e, f, $n = 9$, s.d., Supplementary Fig. 18).

## Discussion

Among many reasons that are limiting the adoption of visual prostheses, invasive implants and subpar vision quality are major concerns. In this study, we described a completely noninvasive methodology for vision restoration, utilizing array-based ultrasound retina stimulation, herein referred to as U-RP. We demonstrated that U-RP stimulated the retina with specific patterns and generate matching patterns in the higher visual pathway. Compared to invasive prostheses, this noninvasive U-RP could alleviate safety concerns and surgery burden, and improve the artificial vision quality, thereby enhancing acceptance rates. Specifically, we demonstrated a spatial resolution better than 100 μm, which correlates to a Snellen visual acuity of 20/400. This surpasses the commercially available Argus II prosthesis (525 μm)[6], is comparable to the clinically tested PRIMA (110 μm)[24] and POLYRETINA[9], and is lower than the recently proposed photovoltaic implant (40 μm)[25]. Besides resolution, FOV is another critical factor that determines the quality of artificial vision. Owing to its full-size FOV, U-RP can generate more informative patterns with a pixel count that is an order of magnitude higher (Supplementary Fig. 19). Large FOV of U-RP also broadens the indications of retina prostheses. Progressive retina diseases can initiate with peripheral photoreceptor loss while the central retina remains intact[26]. Electrode implants are not effective in this case due to the progressively degenerating area and risks associated with large-size implants. In contrast, the noninvasive U-RP, which is inherently effective in stimulating both central and peripheral retina, offers a superior alternative.

The temporal resolution of U-RP is 15 Hz, which is comparable to the electrode-based prosthesis (20 Hz) and sonogenetics (13 Hz)[11], and lower than the fastest optogenetics (>40 Hz)[27]. While a 15-Hz artificial vision might be practically sufficient for patients, it has been reported that an optimized stimulation waveform can enhance efficiency and increase temporal resolution[28,29]. Therefore, future studies in optimizing ultrasound stimulation waveform could potentially improve the performance of U-RP.

A key feature of U-RP is the integration of real-time 3D imaging and automatic alignment. Imaging guidance is the most common tool to ensure precise treatment. Prior studies usually used MRI or other imaging probes to guide the treatment and manually aligned the focus to the targeted area[14,19]. However, this approach is not feasible for personal or daily use. In this work, we carefully balanced the power efficiency and imaging resolution during the design and fabrication of ultrasound 2D arrays to enable real-time imaging-guided stimulation on the retina. This method has the potential to be adapted to stimulate other tissues in the future.

Safety is another important factor in assessing the application potential of U-RP. Although U-RP avoids the need for invasive implants and gene engineering, the ultrasound wave itself could cause damage through negative pressure cavitation or heat deposition. The U.S. Food and Drug Administration (FDA) sets stringent requirements for ophthalmological ultrasound imaging because most eyeball structures (cornea, lens, etc.) lack blood flow, the major source for heat dissipation within the eyeball[30,31]. Ultrasound parameters used in our study were within the FDA requirements for eyes in the high-frequency range, yet higher than the FDA requirement in the low-frequency range (Supplementary Fig. 20). While the ultrasound pressures used in this study were also reported to permeabilize cells and open blood-retina barrier by other works, it should be noted that the barrier opening phenomena require the injection of microbubble agents, which were not used in this study. However, our safety study provided comprehensive evidence proving that either low or high-frequency ultrasound used in our experiment were safe (Supplementary Fig. 20). It is important to distinguish between unfocused imaging fields and focused stimulation fields when assessing the safety of ultrasound retina stimulation. Since ultrasound power was focused on the retina, which has a sufficient distribution of blood vessels, a considerable amount of resulting heat deposition was dissipated. We calculated the Temperature Index (TI) in the eyeball, a standard method to quantify the thermal effect of ultrasound[32], which is lower than 0.3 and within the suggested safety range (Supplementary Fig. 20d).

Although our investigation has proved ARF is the physical mechanism of ultrasound retina stimulation, the biological mechanism still needs further exploration. We hypothesize that ARF activated the remaining RGCs in PD retinas through intrinsic mechanosensitive ion channels. Several types of mechanosensitive ion channels have been observed in healthy retinas[33]. However, no study has been performed on the PD retinas. Here, we examined genes in the RGCs from PD retinas and found broad expression of mechanosensitive ion channels (Supplementary Fig. 21), which explained why PD retinas were sensitive to ultrasound stimulation. Further investigations are still required to identify the role of each ion channel in ultrasound retina stimulation.

While we have shown in this study that U-RP is both effective and safe, the challenge of making it wearable for everyday use remains. Conventional ultrasound probes are typically bulky and handheld, which hinders their applications for daily and continuous use. Fortunately, recent advancements in wearable ultrasound devices, such as bioadhesive ultrasound patches[34] and stretchable ultrasound arrays[35], have shed light on this issue. A curved and thin ultrasound array could be worn like a contact lens on the eye surface and stimulate the retina. Work in this direction is already underway intending to overcome the wearability issue of U-RP[36].

This work demonstrated the practical viability of U-RP and reveals its promising performance and potential applications. By investigating the performance, efficiency, and safety of U-RP, we have paved the path for U-RP from rodent animal research to human study. Future

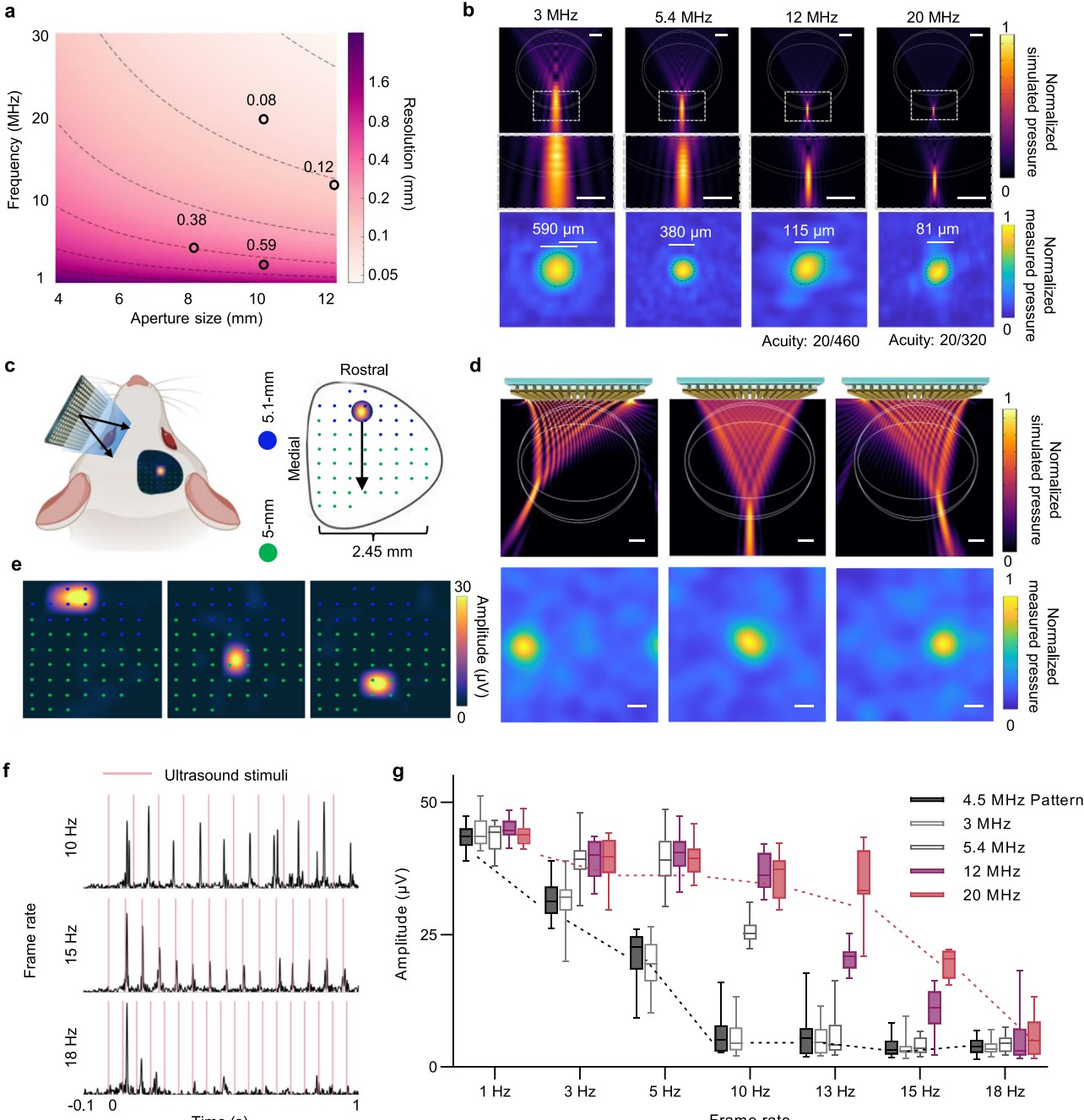

**Fig. 4 | Frequency-dependent spatial-temporal resolution and efficiency of ultrasound retina stimulation. a** The lateral resolution of ultrasound focus, dependent on aperture size and center frequency. The black circles showed the measured resolution of four ultrasound probes at 3, 5.4, 12 and 20 MHz, respectively. **b** Top: Simulated ultrasound fields at different center frequencies in x-z plane. Middle: Zoom-in view of ultrasound fields circled by the dotted white boxes in top panel. Bot: Hydrophone-measured ultrasound fields in x-y plane. **c** Schematic showing that U-RP electrically scan from the nasal side to the temporal side, and the response on SC moves from the anterior side to the posterior side. **d** Top: the simulated scanning 4.5-MHz ultrasound fields in x-z plane. Bot: the hydrophone-measured scanning ultrasound fields in x-y plane at the focal length. The ultrasound focus was steered to −3 mm, 0 mm and 1.5 mm in x direction, demonstrating the ability to cover the whole retina. **e** The mapped neuron responses on the surface of SC. Response moved in according to the ultrasound field scanning. The neuron response movement also covered the whole visual reception field of SC. **f** The temporal resolution of U-RP at 12 MHz: Ultrasound-induced visual signals at different frame rates (12 MHz, 10 ms, 1.6 MPa). **g** Amplitudes of repetitive responses evoked by ultrasound at different center frequencies and frame rates. $n = 8$ animals; The box extends from the 25th to 75th percentiles and the line in the middle of the box is plotted at the median. Whiskers cover from the minimum to maximum. Unless labeled otherwise, white bars indicate the length of 1 mm. Source data are provided as a Source Data file.

studies are desired to develop optimized and wearable stimulation devices and evaluate the quality of regenerated vision in humans. The strategy of using a single 2D array for simultaneous and real-time imaging guidance with dynamic beam steering could also benefit ultrasound neurostimulations on other organs.

## Methods

### Animals

The in vivo rat experiments were performed according to standard ethical guidelines and were approved by the University of Southern California (USC) Institutional Animal Care and Use Committee (IACUC)

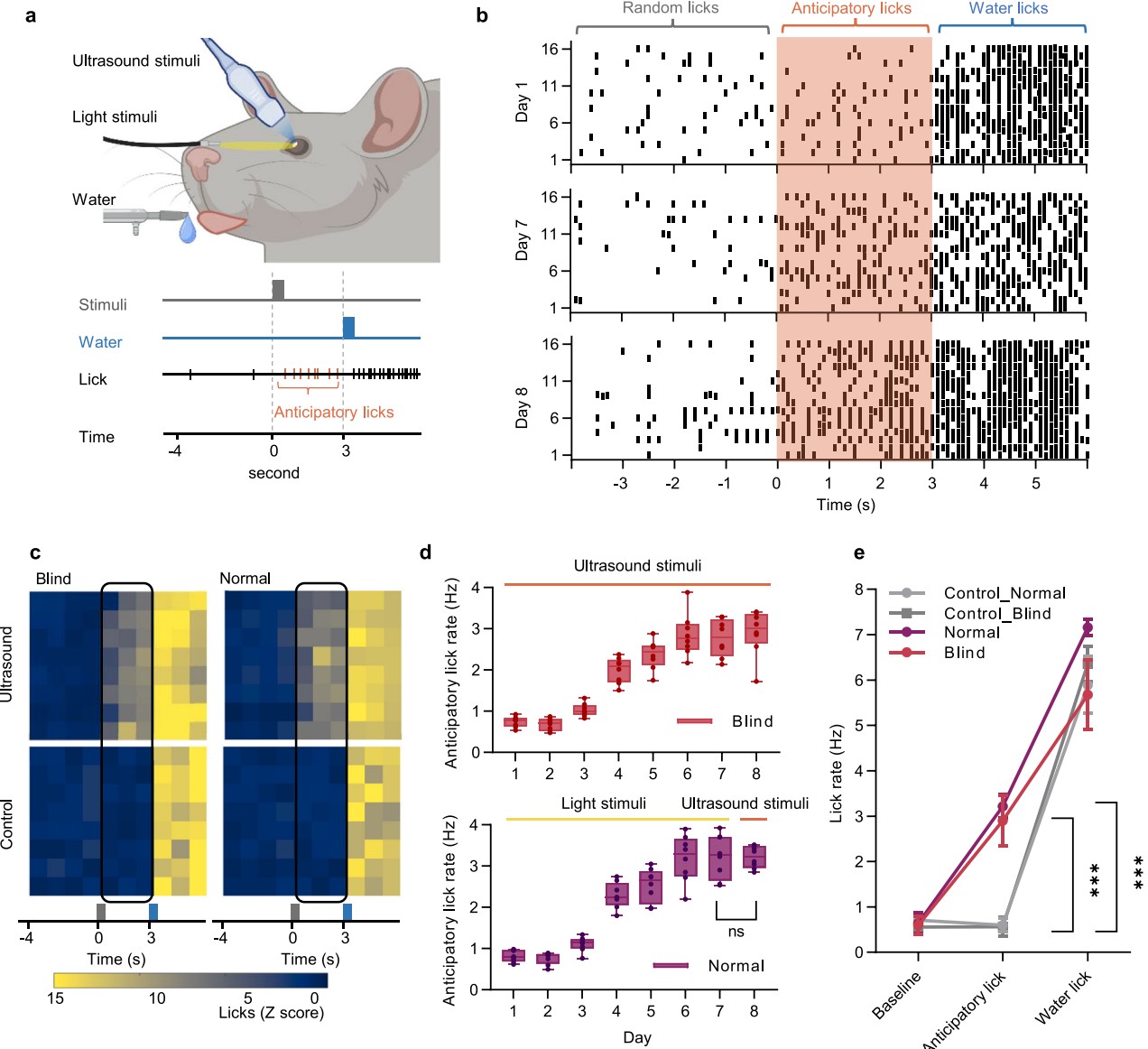

**Fig. 5 | Behavior response induced by the ultrasound retina stimulation.**
**a** Experiment setup of water-licking behavior tests. Light or ultrasound stimulation of one eye cues water drops. Animals responded by licking before (anticipatory licks) or after the appearance of water. **b** Licking responses during representative sessions of a blind rat. Day 1: start of training with ultrasound stimulation. Day 7: end of training with ultrasound stimulation. Day 8: test with ultrasound stimulation. (Ultrasound parameters in both training and testing sessions: 4.5 MHz, 20 ms, 2.9 MPa). **c** Lick response heatmaps recorded in Day 8. Rows: responses of different animals. Eight blind rats and eight healthy rats were used. Columns: responses in one-second time bins. Top: ultrasound stimulation group. Bottom: control group with sham stimulation behind the eye. Left: blind rats. Right: healthy rats. **d** Average

anticipatory lick rates as along the number of training sessions. Top: the results from blind rats that were stimulated by ultrasound in both training and testing sessions. Bot: the results from healthy rats that were trained with light stimulation and tested with ultrasound stimulation. $n = 8$ animals; The box extends from the 25th to 75th percentiles and the line in the middle of the box is plotted at the median. Whiskers cover from the minimum to maximum and all data points. Two sided t test, $p = 0.9889$ (ns). **e** Overall analysis of Mean lick rates in different stages on Day 8. $n = 8$ animals; Mean, s.e.m.; $p = 0.002$, $0.0018$. Ordinary one-way ANOVA and Dunnett's multiple comparison tests were performed to compare the results versus Control_Normal group. Source data are provided as a Source Data file.

protocol 21084. Healthy rats (strain: Long-Evans, stock code: 006) were obtained from Charles River Laboratories. Blind rats with photoreceptors degeneration (strain: Royal College of Surgeon) were bred locally at USC. All rats were male and over 16 weeks of age. The photoreceptor degeneration resulted from retinal pigment epithelium dysfunction, which arose from the deletion of the Mer tyrosine kinase receptor, thereby inhibiting the internalization of photoreceptor outer segments by RPE cells. For each rat, one eye was arbitrarily selected for stimulation, leaving the other eye untreated. Three blind rats and three healthy rats were used in the initial validation of both light and ultrasound stimulation. Eight blind rats were used to investigate the

frequency-dependent acoustic threshold and spatial-temporal resolution investigation, and pattern generation. Eight blind and eight healthy rats were used in the water-licking behavior tests. Two blind rats and two healthy rats were used in the in vivo fiber photometry experiment. Three healthy rats were used in the safety examination study. The unstimulated eye was used in the negative control group. One healthy rat was used in the positive control group.

### Fabrication of ultrasound transducers
To test the frequency-dependent performance of ultrasound retina stimulation, four single-channel ultrasound transducers with focal

lengths of 10 mm were fabricated at 3, 5.4, 12, and 20 MHz. Using the KLM model simulation tool (Biosono Inc., Fremont, CA, USA), the transducers were designed and optimized. Their respective aperture diameters were 10, 8, 12, and 10 mm. Due to its high electromechanical coupling factor and sustainable high-power capability, DL-47 (Del-Piezo Specialties, FL, USA) was employed for the 3-MHz and 5.4-MHz transducers as the piezoelectric layer. For the 12-MHz and 20-MHz transducers, Lithium Niobate (36° Rotated Y-cut LNO, Boston Piezo Optics Inc., Bellingham, MA, USA) was utilized, as LNO maintains a low dielectric constant, ensuring electrical impedance match within higher frequency ranges. For protection and insulation, a layer of 10-μm parylene was coated on the surface of the transducers. The electrical impedance of the fabricated 2D array was assessed in air using an impedance analyzer (E4990A, Keysight, Santa Rosa, CA, USA). The transducers' acoustic fields, both in terms of resolution and absolute pressure value, were measured and calibrated using a hydrophone system (HGL-0085, ONDA Co, Sunnyvale, CA, USA).

### Fabrication of the ultrasound 2D array

The ultrasound 2D array employed in this study encompassed 256 (16 by 16) channels, had a central frequency of 4.5 MHz, and a pitch size of 0.75 mm. Design and material selection were optimized using the KLM model simulation tool (Biosono Inc., Fremont, CA, USA) and summed in Supplementary Table S1. Specifically, DL-47 (Del-Piezo Specialties, FL, USA) was chosen as the piezoelectric layer, due to its high electromechanical coupling factor and high dielectric constant which optimized the power efficiency of the designed array. The kerf between elements was filled with EPO-TEK 301 (Epoxy Technology, Billerica, MA, USA). Considering the array was designed for both imaging and stimulation, the key challenge was to optimize the trade-off between power efficiency and axial resolution. In our design, a 5-mm-thick backing layer made of 3022 E-Solder conductive adhesive (Von Roll, Breitenbach, Switzerland), approximately 12 times the wavelength, was added to the back of the piezoelectric layer. This layer provided mechanical adhesion (adhesive strength: 2030 psi) and electrical connection to the printed circuit board. In addition, due to the relatively low longitudinal sound velocity (1920 m/s) and high acoustic attenuation coefficient (-3.67 dB/mm/MHz)[37], this backing layer improved the axial resolution without a significant increase in the size of the array. A 2−3-μm silver epoxy (silver powder: Fisher Scientific, Hampton, NH, USA; Epoxy: EPO-TEK 301) composite with a thickness of quarter wavelength was attached to the surface of the piezoelectric layer to further enhance the pressure amplitude and axial resolution. Afterwards, a thin layer of parylene C was coated on the array surface for insulation and protection. More details about ultrasound 2D array design and fabrication can be found in the previous work[38]. The performance of each element in the fabricated 2D array (center frequency, bandwidth, signal amplitude, and level of crosstalk) was characterized by connecting the array to the 256-channel Vantage system (Verasonics, Inc., Kirkland, WA, USA). The pulse-echo signal (1-cycle pulse, 20 Vpp) of each element was acquired and analyzed to determine the center frequency, bandwidth, and signal amplitude. To quantify the level of crosstalk, one element was excited (100 cycle at 4.5 MHz, 30 Vpp), then the voltages across the nearest adjacent element were measured and compared with the voltage measured in the excited element.

### Imaging-guided pattern stimulation via ultrasound 2D array

**Imaging.** Radio-frequency signals from the ultrasound 2D array were acquired and processed by the 256-channel Vantage system with an extended transmit upgrade (Verasonics, Inc., Kirkland, WA, USA). A customized three-point-focus line mode scan 3D imaging sequence (adapted from the algorithm packages in the Verasonics system) was developed to optimize the imaging quality, particularly for the surface and the bottom of the eyeball. The visualization of 3D reconstruction

was conducted using Amira 2019 (Thermo Fisher Scientific, Waltham, MA, USA).

**Auto-alignment.** Auto-alignment was enabled by automatically detecting the relative depth and angle of the eyeball to the surface of the 2D array. To detect the cornea surface and retina layer, edge detection was performed using the canny method in MATLAB. The depth was determined by the bottom edge, while the angle was determined by the normal direction of the top edge surface.

**Pattern generation.** By controlling the phase and amplitude in each element, an Ultrasound 2D array can generate arbitrary patterns at desired locations. We used the band-limited angular spectrum method[39] to compute the phase and amplitude distributions, as it yields accurate results and has a low computational cost. The angular spectrum of the acoustic pressure wave in a plane at constant depth $z$ can be expressed in the spatial frequency domain:

$$P\left(k_x, k_y, z\right) = \int\int p(x, y, z) e^{-j(k_x x + k_y y)} dx dy \tag{1}$$

We define $z = 0$ to be the array surface. Once P(kx, ky, 0) is known, the angular spectrum at any plane downstream can be calculated by multiplying the angular spectrum with a propagator function:

$$P\left(k_x, k_y, z\right) = P\left(k_x, k_y, 0\right) e^{jz\sqrt{k_m^2 - k_x^2 - k_y^2}} \tag{2}$$

where $k_m = |\mathbf{k}|$ is the wavenumber in the liquid medium, and the wave vector $\mathbf{k} = (kx, ky, kz)$. The real-space pressure field in any plane z can be obtained by performing the inverse Fourier transform. Back-propagation from the image to the hologram can be calculated with:

$$P\left(k_x, k_y, z\right) = P\left(k_x, k_y, 0\right) e^{-jz\sqrt{k_m^2 - k_x^2 - k_y^2}} \tag{3}$$

The limited size of the array surface leads to a cut-off of higher spatial frequencies over the propagation distance. Therefore, the integration region of the inverse Fourier transform was limited to account for the cut-offs:

$$k_{x,y} \leq \frac{\pi\left(L_a + L_i\right)}{\lambda\sqrt{z^2 + \frac{(L_a + L_i)^2}{4}}} \tag{4}$$

where $\lambda$ is the wavelength in the medium, and $L_a$, $L_i$ are the side lengths of the array and the targeted pattern plane.

If the auto-aliment suggested an angle deviation ($\theta_x$, $\theta_y$) between the array and eyeball, this deviation would be compensated by adding an additional phase distribution:

$$\varnothing(x, y) = k_m\left(x \tan\theta_x + y \tan\theta_y\right) \tag{5}$$

where $\varnothing(x, y)$ is the phase distribution in the range of (0, 2π).

After computation, the phase and amplitude distributions were applied to the array via the 256-channel Vantage system with extended transmit upgrade (Verasonics, Inc., Kirkland, WA, USA). The pattern generation was confirmed by doing hydrophone mapping.

Image data processing, auto-alignment, pattern generation and Vantage system control were integrated in customized scripts operated by the MATLAB 2019b (MathWorks, Natick, MA, USA) and the overall running time is within 50 ms on the test platform (Intel i7-7700K@3.60 GHz).

### In vivo electrophysiological recording

Before the experiment, rats were anesthetized via an intraperitoneal injection of Ketamine/Xylazine (50-90 mg/kg, 5-10 mg/kg), head-fixed

in a stereotaxic frame, and maintained under anesthesia using sevoflurane inhalation through a nose cone. Ophthalmic ointment was applied to both eyes to prevent dryness. The animal's body temperature was regulated with a heating pad. A circular craniotomy of approximately 3.5 mm in diameter was performed above the primary visual cortex. After the skull flap was removed, the exposed cortical surface was maintained in a moist condition using a cortex buffer that contained 125 mM NaCl, 5 mM KCl, 10 mM glucose, 10 mM HEPES, 2 mM MgSO4, and 2 mM CaCl2.

Neural activity from the Superior Colliculus (SC) was recorded using a customized 56-channel multielectrode array (MEA) with an impedance of 0.5 Mega-Ohms and a tip-to-tip spacing of 350 μm. This MEA was custom designed to cover the surface of the SC (Microprobes for Life Science, Gaithersburg, MD, USA). Electrode lengths were varied to accommodate the curvature of the SC (Fig. 3b). The signals from the MEA were sampled at a rate of 30 kHz by two analog-to-digital multiplexing headstages (HS−32-MUX-PTB, Neuralynx, Bozeman, MT, USA) before being transferred to the 64-channel Lablynx recording system (Neuralynx, Bozeman, MT, USA). The MEA was carefully advanced into the SC gradually using the stereotactic apparatus. The precise placement of the MEA was confirmed through stable visual responses. To ensure the sensitivity of the retina, all procedures were carried out in a dark room illuminated with dim red light. Rats were euthanized while under anesthesia at the end of the recording procedures.

### In vivo fiber photometry experiment
**Virus injection.** Rats were first anesthetized with an intraperitoneal injection of Ketamine/Xylazine (50-90 and 5-10 mg/kg), followed by sevoflurane inhalation (2−2.5%) through a nose cone to ensure a deep level of unconsciousness. A hole with a 2 mm diameter was drilled above the primary visual cortex. The GCamp8 virus (pGP-AAV-syn-jGCaMP8m-WPRE, titer ≥$1 \times 10^{13}$ vg/mL, Addgene, Watertown, MA, USA) was then injected into the brain at an anteroposterior (AP) coordinate of −5.56 mm, mediolateral (ML) coordinate of +3.64 mm, and dorsoventral (DV) coordinate of −1.5 mm using the RWD R-480 microinjector. This injection process lasted for 30 min, after which the needles were left in place for an additional 8 min to facilitate the diffusion of the virus. The needles were then raised by 1 mm and left in place for 2 additional minutes to further promote virus diffusion and minimize its spread along the injection tract.

**Fiber-optic cannula implant.** Following the virus injection, a fiber-optic cannula (Fiber optic pins, RWD, R-FOC-BL200C−39NA, with D = 1.25 mm Ceramic Ferrule, 200 μm Core) was implanted through the same site. To protect the dura, a Silicon Elastomer (World Precision Instruments) was used. Three additional holes were drilled near the fiber-optic cannula site into which three skull nails were inserted. The cannula and the skull nails were then secured together using dental cement.

**Fiber photometry system.** The fiber photometry system (Neuroscience Console 500 and Integrated Flourance Mini Gen3, Doric Lense, Canada) was utilized to emit a 470 nm excitation light and collect fluorescence signals, selectively recording responses from the primary visual cortex.

### Ultrasound stimulation system
During the experiment, the ultrasound transducer was precisely positioned using a 5-axis precision stage to accurately control its position and angle. A degassed ultrasound gel served as a coupling medium between the transducer surface and the rat's eye. A dual-channel function generator (AFG3252C, Tektronix, Beaverton, OR, USA) was utilized to manage the stimulus sequence and trigger signals for data acquisition. Specifically, the output of channel 1 was employed to generate the stimulus sequence, which was then amplified by an RF power amplifier (100A250A, Amplifier Research, Souderton, PA, USA) with a gain of 50 dB, and used to operate custom-built ultrasound transducers. Channel 2, which is internally synchronized with channel 1, sent synchronized trigger signals to the interface board of the Lablynx electrode recording system (Neuralynx, Bozeman, MT, USA). To minimize artifacts, the stimulation system and the recording system were grounded together.

### Ultrasound stimulation sequence and parameters
Unless otherwise specified, the default US sequences used in this study were 10-ms pulse with 100% duty cycle. The frame interval was at least 6 seconds to ensure the stimulated neurons fully recovered after each stimulation. The acoustic pressure is referred to as the negative peak pressure (NPP). The mechanical index (MI) was calculated as $MI = \frac{NPP(MPa)}{\sqrt{f(MHz)}}$, where $f$ was the center frequency. The spatial peak pulse average intensity ($I_{SPPA}$) and spatial peak temporal average intensity ($I_{SPTA}$) were calculated as: $I_{SPPA} = \frac{NPP^2}{2\rho c}$, $I_{SPTA} = I_{SPPA} \times Dutycycle$, where $\rho$ and $c$ are the density and sound speed in the medium. The thermal index (TI) at the retina was calculated as $TI = \frac{W_p}{W_{deg}} = \frac{min(W, I_{spta} \cdot 1cm^2)}{210mW \cdot MHz}$, where $W_p$ is the derated acoustic power at the depth of interest and $W_{deg}$ is the estimated power necessary to raise the tissue equilibrium temperature by 1 °C according to a selected specific tissue model.

### Light stimuli
A full-field strobe flash using a Grass Photic stimulator (Grass Instrument Co., West Warwick, RI, USA) was delivered to the eye. In the meanwhile, the stimulator sends out a trigger signal to the Lablynx recording system for data recording. The time interval between each adjacent trigger signal is set to at least 6 seconds in order to ensure the retinal neuron activities are back to normal.

### Behavior test
Before the experiment, the rats underwent three days of water deprivation, with weight loss maintained between 80 and 85%. Animals were gradually acclimatized to the experimental setup, including head and body fixation, along with water reward. The rat's head was held in place using an ear bar while the body was secured within a cylindrical acrylic tube with tape to ensure maximum comfort during the head fixation. Healthy rats were initially trained to associate light stimulation with water rewards. A 20-ms pointed white LED light was projected into the left eye of each rat during the experiment. This was followed by two water drops dispensed through a blunt 18-gauge needle situated 6 mm away from the mouth, three seconds after the light stimulation. Inter-trial intervals were randomly set between 60 and 120 s, with typical sessions lasting approximately 30 min during which rats performed 16–24 trials. This training session was repeated over seven consecutive days. On the eighth day, the light stimulation was replaced with ultrasound stimulation. Ultrasound stimulation was administered for one day. In a subsequent experiment, blind rats were tested under identical procedures and parameters, with the sole difference being the direct application of ultrasound stimulation. This test was performed over a consecutive eight-day period. Lick events were detected via a transistor-based circuit and recorded by a USB 6002 (National Instruments, Austin, TX, USA) data acquisition device. For Z scores, a 1-second background time interval was used to calculate mean and standard deviation. The experiment setup is shown in Supplementary Fig. 16. To prevent noise from equipment and devices from affecting the animals' behavior, the animal table was surrounded by soundproofing acoustic foam (Acoustical Surfaces, Chaska, MN, USA).

## ARF equation fitting

ARF is a non-linear physical phenomenon occurring in acoustic wave propagation. As acoustic waves are reflected and attenuated within media, the momentum of the waves, which moves in the direction of wave propagation, is transferred to the media (in the case of medical ultrasound, this refers to biological tissues), applying a non-contact, unidirectional mechanical force. It is calculated as follows[40]:

$$F = \frac{p^2 \cdot \alpha f}{\rho c^2} \qquad (6)$$

where F is the force per unit volume, $\alpha$ is the attenuation coefficient, $f$ is the center frequency, $p$ is acoustic pressure, $c$ is the sound speed, and $\rho$ is the density. Fitting experiment results to the equation were performed using the curve fitting function in MATLAB 2019b (MathWorks, Natick, MA, USA).

## In vivo safety tests

This study incorporated a systematic evaluation of the long-term safety associated with ultrasound retina stimulation. We subjected three healthy rats, each with one eye under ultrasound stimulation at frequencies of 3, 4.5, and 20 MHz. The pulses had a duration of 10 ms and a repetition frequency of 5 Hz, with pressure amplitudes surpassing the threshold by 50% (derated pressures were 5.8 MPa, 4.3 MPa, and 1.7 MPa). In the positive control group, the 20-MHz ultrasound were applied for three hours with a pressure 100% higher than the safe intensity (3.5 MPa vs 1.7 MPa, 10-ms pulse duration, and a repetition frequency of 5 Hz), in order to elicit an inflammatory response in the retina. Each rat underwent stimulation for 3 hours tri-weekly for two weeks (Day 1 to Day 13). The rats were anesthetized with a Ketamine/Xylazine cocktail (50-90 mg/kg, 5-10 mg/kg), head-fixed in a stereotaxic frame, and kept under sevoflurane inhalation without requiring surgery. Prior to and after the stimulation period, we carried out eye examinations (OCT and fundus imaging, ERG measurement) on both eyes (Day 0 and Day 14). Following these assessments on Day 14, the rats were euthanized under anesthesia, with both eyes collected for histological examination.

**OCT and fundus imaging.** After anesthetization in the isoflurane induction chamber, the rats were carefully transferred to the experiment table, and their body temperatures were maintained with a heating pad. Pupil dilation was achieved using tropicamide eye drops, and after a waiting period of 10–15 min, any debris was removed using sterile cotton swabs. A sterile saline solution was used to keep the eyes moist. The OCT and fundus imaging system (HRA + OCT PECTRALIS, Heidelberg Engineering Inc., Franklin, MA, USA) was aligned and focused to capture images of the retina.

**ERG measurement.** After anesthetization in the isoflurane induction chamber, the rats were carefully transferred to the experiment table, and their body temperatures were maintained with a heating pad. Pupil dilation was achieved using tropicamide eye drops, and after a waiting period of 10–15 min, any debris was removed using sterile cotton swabs. A sterile saline solution was used to keep the eyes moist. The OCT and fundus imaging system (HRA + OCT PECTRALIS, Heidelberg Engineering Inc., Franklin, MA, USA) was aligned and focused to capture images of the retina.

## Histology analysis

Following the retina stimulation experiments, the eyes were either immersed in Bouin's fixative or embedded in paraffin after the rats were euthanized. Transverse sections of the retina were then cut, mounted onto slides, and stained with hematoxylin-eosin (H&E).

**GFAP.** For immunostaining, all slides were deparaffinized, rehydrated, and antigen retrieved (sodium citrate, pH 6.0). After that, the sections were incubated in blocking buffer (10% donkey serum, 0.2% Triton X-100 [Sigma-Aldrich, St. Louis, MO, USA] in PBS) for 1 h at room temperature, followed by incubation with primary antibody (GFAP-mAb #3670, (dilution of 100) Cell Signaling Technology (Danvers, MA) at 4 °C overnight. After several phosphate-buffered saline (PBS) washes, slides were incubated for at least 1 hr at room temperature in fluorescent-labeled secondary antibody (Alexa Fluor 488 goat anti-rabbit IgG (H + L) (dilution of 1:300 (Thermo Scientific, USA). After staining, the slides were mounted with a fluorescent-enhanced mounting medium with 4′,6-diamidino-2-phenylindole (DAPI) (Vector Laboratory, Burlingame, CA, USA).

**CD68.** All slides were deparaffinized, rehydrated, and antigen retrieved (sodium citrate, pH 6.0). After that, the sections were incubated in blocking buffer (10% donkey serum, 0.2% Triton X-100 [Sigma-Aldrich, St. Louis, MO, USA] in PBS) for 1 h at room temperature, followed by incubation with primary antibody (CD68, PIMA513324, Mouse anti-Rat, CD68 Monoclonal Antibody (KP1), Invitrogen™, USA) at 4 °C overnight. After several phosphate-buffered saline (PBS) washes, slides were incubated for at least 1 hr at room temperature in fluorescent-labeled secondary antibody (DyLight™ 594, Goat anti-Mouse IgG (H + L) Secondary Antibody, PI35510, Invitrogen™, USA). After staining, the slides were mounted with a fluorescent-enhanced mounting medium with 4′,6-diamidino-2-phenylindole (DAPI) (Vector Laboratory, Burlingame, CA, USA). Images were taken using the Ultra viewer ERS dual-spinning disk confocal microscope (PerkinElmer, Waltham MA, USA) equipped with a C-Apochromat (Carl Zeiss, Thornwood, NY, USA) ×10 high dry lens, a C-Apochromat ×40 water immersion lens NA 1.2, an electron multiplier charge-coupled device cooled digital camera (Hamamatsu Orce_ERCC 12-bit camera]; PerkinElmer, Waltham, MA, USA)

## RGC RNA-sequencing

Rats were euthanized using Isoflurane anesthesia followed by cervical dislocation. After removing the eyes, globes were enucleated and retinas were dissected and pooled in Hank's Balanced Salt Solution. Retinas were dissociated using the MACS neural dissociation kit (Miltenyi Biotec, Charlestown, MA, USA) and washed using PBS and incubated with Enzyme Mix 1, containing a low concentration of papain for 15 min with continuous mixing. After adding Enzyme Mix 2, retinas were then dissociated by 10 passages through a serological 5 ml pipette and incubated for a further 10 min in a tube Rotator. Tissue pieces were further dissociated by passaging using a Pasteur pipette after adding the second part of Enzyme Mix 2. After this, a 40 μm filter was used to separate single cells and they were washed by centrifugation.

RGCs were then isolated using the Retinal Ganglion Cell Isolation Kit (RGC Isolation Kit, Miltenyi Biotec) according to the manufacturer's instructions. Briefly, RGCs were incubated for 5 min with CD90.1/Thy1.1-MicroBeads and then for an additional 10 min with a biotinylated depletion antibody. After washing, cells were resuspended in 750 μl of D-PBS/BSA and mixed with anti-Biotin-MACSiBeads (Miltenyi Biotec) for 15 min. Unwanted cells labeled by the biotinylated antibody-MACSiBead complex were then depleted using a MACSiMAG Separator. Non-depleted RGCs labeled with CD90.1 MicroBeads were subsequently enriched using two MS columns and an OctoMACS Separator. Cells were then eluted from the second MS column and resuspended in serum-free MACS Neuro Medium (Miltenyi Biotec) supplemented with 50 ng/mL brain-derived neurotrophic factor (BNDF) (Peprotech, Rocky Hill, NJ, USA; 450-02), 50 ng/mL ciliary neurotrophic factor (CTNF) (Peprotech, 450-13), and 5 μM forskolin (StemCell Technologies, Cambridge, MA, USA; 72114). Purified RGCs were quantified and then diluted to $0.5 \times 10^6$ RGCs/mL in the medium. Cells (1 mL) were then plated onto a 35 mm dish and placed in an incubator at 37 °C overnight for 12 h.

Then we extract the RNA from cells. The first part is sample disruption and homogenization. Cell pellets were procured by centrifugation at $500 \times g$ for 1 min. The supernatant was carefully discarded, and the pellet was gently resuspended in 300 μL of RNA lysis Buffer to prevent foaming. The next part is RNA Binding and Elution. A maximum of 800 μL of the sample from Part 1 was transferred to a gDNA removal column equipped with a collection tube. A 30-second centrifugation was performed to eliminate the majority of gDNA, and the flow-through was retained. An equal volume of ethanol ($\geq 95\%$) was added to the flow-through, followed by thorough mixing. The mixture was then transferred to an RNA purification column with a collection tube and centrifuged for 30 seconds. The column was sequentially washed with 500 μL RNA Priming Buffer and 500 μL RNA wash buffer, each followed by a 30-s centrifugation. Another wash step was performed with 500 μL RNA wash buffer, followed by a 2-min centrifugation. The column was then transferred to an RNase-free microfuge tube. Between 30 and 100 μL of Nuclease-free water was added directly to the center of the column matrix, followed by a 30-second centrifugation. The RNA was stored at −20°C for short-term preservation. Then the RNA sample was dispatched to Azenta (Chelmsford, MA, USA) for comprehensive analysis.

### FEM simulation

The finite element method (FEM) simulation was conducted using COMSOL Multiphysics 6.0 (Stockholm, Sweden). The Acoustic module and the Bioheat transfer module were coupled for simulations of the acoustic field and ultrasound-induced tissue thermal effects in this study. Both frequency-domain studies and time-dependent studies were performed. In the simulation setup, the eyeball was simplified into four main parts: the cornea, lens, vitreous body, and retina, where the shape and size of each part were preset. The acoustic and thermodynamic properties of each part were determined based on previous literature[41]. Detailed parameters are listed in Supplementary Table S2.

### Ultrasound-induced thermal effect in rat eyeball ex vivo

Two rat eyeballs were used to estimate the ultrasound-induced temperature increase. In regard to the US attenuation measurement, the hydrophone was placed at the focal point of the transducer (10 mm in depth). The intact eyeball was held by a clip. The acoustic pressure was measured three times, both with and without the presence of the eyeball. As for the temperature measurement, a T-type wire thermocouple (XC-T-TC-WIRE, Omega Engineering Inc., CT, USA) was inserted into the posterior eye, and the temperature was read out from a thermocouple meter (RH820U, Omega Engineering Inc., CT, USA).

### Electrophysiological data post-processing

**Signal processing.** In experiments, raw signals picked up by electrodes were sampled at 30 kHz and stored for offline post-processing in MATLAB 2019b (MathWorks, Natick, MA, USA). Step-by-step processing is illustrated in Supplementary Fig. 22. To quantify the amplitude of multi-unit neuron activities (MUA), raw signals were first filtered by a 500–7000 Hz bandpass filter, then rectified and filtered by a 10–200 Hz bandpass filter. The amplitude of the signal was determined by the maximal peak value. A subpopulation of MEA channels (usually one or two channels) with strong background noise (presumably because the electrodes were inserted near blood vessels) was manually set to zero.

**SC mapping.** The amplitude distribution of MUA from all 56 MEA channels was plotted to map the neuron responses at SC. For better visualization, a 4-times modified Akima cubic 2D spatial interpolation of amplitude distribution was conducted. Interpolation and figure representation were performed using MATLAB 2019b (MathWorks, Natick, MA, USA). The structural similarity index was calculated using the SSIM function in MATLAB 2019b.

### Statistical analysis

Statistical analysis and graphical representation of three or more datasets were conducted by the Prism 9 software (GraphPad).

### Reporting summary

Further information on research design is available in the Nature Portfolio Reporting Summary linked to this article.

## Data availability

A sample data of raw neuron activities in this study have been deposited in the https://figshare.com/articles/dataset/UltrasoundRP/25438252. All data generated in this study are provided in the Supplementary Information/Source Data file. Any additional requests for information can be directed to, and will be fulfilled by, the corresponding authors. Source data are provided with this paper.

## Code availability

The scripts for neuron signal processing and visualization, and the generation and validation of arbitrary acoustic pattern used in this study have been deposited in Github at https://github.com/gengxilu/UltrasoundRP.git (https://doi.org/10.5281/zenodo.10989656)[42] with instructions. The ultrasound-control scripts and edge detection scripts were adapted directly from built-in MATLAB functions and Verasonics scripts.

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

## Acknowledgements

This work was supported by the National Institutes of Health (NIH) under grant R01EY032229, R01EY028662, R01EY030126 (Q.Z.). Unrestricted Grant to the Department of Ophthalmology from Research to Prevent Blindness, New York, NY (M.S.H). Research reported in this publication was also supported by the National Eye Institute of the National Institutes of Health under Award Number P30EY029220 (M.S.H). The content is solely the responsibility of the authors and does not necessarily represent the official views of the National Institutes of Health. G. Lu disclosed the support for publication of this work from the Alfred E. Mann Innovation in Engineering Doctoral Fellowship. Parts of Figs. 2–5 and Supplementary Fig. 4 were created with BioRender.com.

## Author contributions

G.L., C.G., and X.Q. designed the experiments. Y.S., H.K., R.L., Y.Z., J.Z. L.J., and G.L. carried out the design, fabrication and characterization of ultrasound probes and systems. C.G., J.J., D.S.R., J.C., C.C., B.T., and G.L. designed and conducted in vivo and ex vivo animal experiments. G.L. and C.G. performed data analysis and simulations. G.L., X.Q., B.T., M.S.H., and Q.Z. conceived the idea for this project. B.T., M.S.H., and Q.Z. supervised the analysis of the obtained data. G.L. and C.G. wrote the manuscript. All the authors provided critical feedback on the research and the manuscript.

## Competing interests

The authors have filed for a patent in U.S. (PCT/US22/13163) for devices and methods described in this work.
