## [Peer Review file · Nature Communications]

REVIEWER COMMENTS

Reviewer #1 (Remarks to the Author):

This paper proposes a new artificial vision system that induces phosphene by ultrasound stimulation of the retina from outside the body and demonstrates the effectiveness of the system using rats. The paper shows excellent results in that the authors actually stimulated the rat retina from outside the eye using the fabricated 2D ultrasound array and recorded the neural responses corresponding to the stimulation in the superior colliculus as a 2D pattern using MEA (fig. 3f). The effectiveness of this method is convincingly demonstrated by conducting not only neural responses but also behavioral experiments.

However, the following points should be clarified to make the device more convincing as a device for human application.

1. The method of performing 3D retina imaging using a 2D ultrasound array for retinal stimulation and aligning the ultrasound stimulation based on the imaging results is clever. However, in practice, it takes time to process the pattern generation algorithm from the acquired images and to perform spatial correction feedback, which makes the reviewer wonder if the positioning can follow normal eye movement. The authors should explain this point.
2. All experiments in this paper were performed on rats. The size of the human eyeball is much larger than that of the rat eyeball, so the ultrasonic waves reach the retina over a longer distance. Since the attenuation of ultrasonic waves increases with increasing frequency, the attenuation may not be negligible at high resolution. The generation of heat is also a concern. Consideration is needed regarding the appropriate frequency and spatial resolution that can be expected to be achieved when this device is applied to actual humans.

Reviewer #2 (Remarks to the Author):

The presented study by Lu and colleagues on noninvasive ultrasonic stimulation in the context of vision restoration is quite intriguing. The authors utilize an ultrasound-based approach to stimulate the retinae of rats. This is paired with MEA recordings in the superior colliculus and a behavioral assay of light-stimulated anticipatory licking. In addition, the extended data set includes optical recordings of calcium transients in visual cortex. The results beautifully demonstrate that ultrasonic stimulation of the retina is able to produce signals in the superior colliculus that correlate with the presented stimuli on the piezo-ultrasound device. In addition, the behavioral data suggest that these ultrasound stimuli, when directed at the eye, are ultimately received and interpreted as visual stimuli. The authors have additionally

addressed some of the health-concerns that may arise from the use of ultrasonic stimulation, although clearly more would need to be done prior to use in humans.

Specific Comments

1. Ultrasound can permeabilize cells, and pressures and durations similar to those used here disrupted the blood-retinal barrier in one 2020 study using focused ultrasound. Have the authors performed an experiment to exclude the possibility of blood-retinal barrier disruption? Is there evidence to rule out extravasation, for example?

2. Microglial also express piezo channels that control their activity. The immunolabeling for microglial cells experiment lacks a positive control. How does the reader know that this antibody worked? One would expect to see microglial in the intact rodent retina plexiform layers, so one could demonstrate that there are microglial cells present and then ask whether numbers or distribution changed, in addition to whether or not they became activated.

3. Figure 5B. it appears that this animal responded to light with anticipatory licks on day 7. However, this animal is also described as being “blind” in the figure legend. Please clarify whether this animal is blind and if so, why it has anticipatory licking in response to light.

4. Figure 5C. Please clarify in the figure legend the day corresponding to the data that is shown?

5. Figure 5D. Earlier in the figure, different sessions across several days are shown. To which sessions or days do these data refer?

6. It is not always clear how many animals were studied for each condition for each set of studies. Please provide this information for each set of experiments. Figure 1, for examples, provides the number of neurons, but not the number of animals. One would like to know that data are reproducible across animals.

7. Supplemental Figure 13 shows changes in gene expression for several proteins. Could these changes be indicative of pathology or a response to a noxious stimulus? What is the significance of the changes in *Itgb3*, for example?

Minor Comments

1. The similarity of the ERG a and b waves in controls versus rats exposed to multiple episodes of ultrasound stimulation is encouraging. However, these waves report activity of photoreceptors and bipolar cells and not retinal ganglion cells. If the ultrasound waves are impacting neurons or cells other than photoreceptors and bipolar, that could not be readily detected by analyzing the a and b waves.
2. Why were all the rats that were used male?
3. Figure 1f – correct spelling of healthy
4. Figure 5E is difficult to visually interpret. The lines connecting each condition detracts from the actual data. Also, what do the grey boxes in between baseline and anticipatory licks represent?

REVIEWER COMMENTS

Reviewer #1 (Remarks to the Author):

General Comment: This paper proposes a new artificial vision system that induces phosphene by ultrasound stimulation of the retina from outside the body and demonstrates the effectiveness of the system using rats. The paper shows excellent results in that the authors actually stimulated the rat retina from outside the eye using the fabricated 2D ultrasound array and recorded the neural responses corresponding to the stimulation in the superior colliculus as a 2D pattern using MEA (fig. 3f). The effectiveness of this method is convincingly demonstrated by conducting not only neural responses but also behavioral experiments. However, the following points should be clarified to make the device more convincing as a device for human application.

Response to General Comment: Thank you very much for acknowledging our work and for your insightful and constructive feedback. Emphasizing the feasibility of ultrasound retina prosthesis for human applications is a key aspect we wish to highlight. We have addressed your comments in a point-by-point manner. Any sentences that have been changed or newly added to the manuscript and the supplementary information are highlighted in blue.

Comment 1: The method of performing 3D retina imaging using a 2D ultrasound array for retinal stimulation and aligning the ultrasound stimulation based on the imaging results is clever. However, in practice, it takes time to process the pattern generation algorithm from the acquired images and to perform spatial correction feedback, which makes the reviewer wonder if the positioning can follow normal eye movement. The authors should explain this point.

Response to Comment 1: Thank you for recognizing the merits of our imaging-guidance method and for bringing up the issue of delays. Tracking eye movement is crucial for accurate retinal stimulation due to the significant changes in eye movement velocity across different modes¹. For example, eyes velocity can be 0°/s for 50-600 ms in the fixation mode², <30°/s in smooth pursuit mode, and as high as 700°/s in saccades mode. There are also involuntary movements (microsaccade, drift, tremor, etc.)³ with a small magnitude of up to 1°.

In our conceptual design for human application, this problem will be naturally resolved from the device side, instead of analyzing the algorithm delay and eye movement speed. We plan to have a wearable ring-shaped ultrasound array that can be worn as contact lenses. In this case, the array will naturally follow any eye movements during daily use. The imaging guidance will be used for calibration when the user puts on the contact-lens-like array, and monitors the array's position from time to time (such as, every five seconds) to avoid accidental errors caused by contact lens shift or fall out. Therefore, tracking fast eye movements is not required in this scenario.

In our conceptual design aimed at human application, the issue of delay will be naturally mitigated by the device itself, rather than relying on analyzing algorithm delay and eye movement speed. We plan to have a wearable ring-shaped ultrasound array⁴ designed to function as contact lenses. This design ensures that the array naturally aligns with any eye movements during daily activities. Consequently, tracking rapid eye movements becomes unnecessary. In this context, imaging guidance will primarily serve for initial calibration when the user wears the contact-lens-like array and will periodically monitor the array's position (e.g., every five seconds) to prevent misalignment or displacement caused by the contact lens shifting or falling out.

However, it's important to highlight that even without the contact-lens-like ultrasound array, the delay is within an acceptable range for tracking eye movements. The overall delay in the retina stimulation system encompasses two components: an algorithm for spatial correction detection and another for generating patterns with spatial correction feedback. Our algorithms operate with a latency of less than 50 ms—less than 35 ms for edge detection and less than 15 ms for pattern generation—on our system (MATLAB, Intel i7-7700K @ 3.60 GHz). This latency results in an angular error of up to 1.5° in smooth pursuit mode and 35° in saccade mode. Considering that involuntary eye movements (less than 1°) are barely noticeable even for healthy individuals, an angular error of 1.5° can be deemed acceptable for patients utilizing visual prostheses. While we acknowledge that the angular error could be significant in saccade mode, it is pertinent to note that patients with acquired blindness—as opposed to congenital blindness, who represent the primary demographic for retina prostheses and have gradually or suddenly lost photoreceptor function in adulthood—are capable of controlling their eye movements similarly to healthy individuals^{5,6}. They can consciously minimize or avoid saccadic movements when utilizing ultrasonic retina prostheses in their daily lives.

In conclusion, normal eye movements will not pose a significant issue for the effectiveness of ultrasonic retina prostheses. The angular error induced by eye movement remains within an acceptable range (1.5°), and the innovative contact lens design further mitigates this concern.

The delay of algorithm and test platform have been added in the **Method - Imaging-guided pattern stimulation via ultrasound 2D array**. Additionally, a new note discussing the error associated with delay has been added to the supplementary information as the **Supplementary Note 1**:

Supplementary Note 1: Eye movement tracking and delay.

Tracking eye movement accurately is critical for precise retinal stimulation, given the significant variability in eye movement velocity across different modes¹. For instance, eye velocity can remain at 0°/s for 50-600 ms in fixation mode², below 30°/s in smooth pursuit mode, and surge to as high as 700°/s in saccades mode. Furthermore, involuntary movements³ like microsaccades, drift, and tremor, though minor (up to 1°), also play a role.

The main contributors to delay in the retina stimulation system are the algorithms for spatial correction detection and pattern generation with spatial correction feedback. Our algorithms demonstrate efficiency, with processing times under 50 ms—less than 35 ms for edge detection and under 15 ms for pattern generation—on our system (MATLAB, Intel i7-7700K @ 3.60 GHz). This results in an angular error of up to 1.5° in smooth pursuit mode and 35° in saccades mode. Given that involuntary eye movements are barely perceptible even in healthy individuals, an angular error of 1.5° should be deemed acceptable for patients using visual prostheses. We acknowledge that the angle error is more pronounced in saccade mode. However, patients with acquired blindness—who represent the primary demographic for retinal prostheses and have typically lost photoreceptor function in adulthood—are capable of controlling their eye movements similarly to sighted individuals. They can consciously minimize or completely avoid saccadic movements while using the ultrasonic retinal prostheses in their daily activities^{4,5}.

In conclusion, standard eye movements are unlikely to pose significant issues for the functionality of ultrasonic retinal prostheses. The acceptable range of angular error (1.5°) and the potential for further improvement with contact lens design ensure that eye movement does not compromise the effectiveness of these devices.

Reference:

- 1 Land, M. & Tatler, B. *Looking and acting: Vision and eye movements in natural behaviour*. (Oxford University Press, 2009).
- 2 Hessels, R. S., Niehorster, D. C., Nyström, M., Andersson, R. & Hooge, I. T. Is the eye-movement field confused about fixations and saccades? A survey among 124 researchers. *Royal Society open science* **5**, 180502 (2018).
- 3 Martinez-Conde, S., Otero-Millan, J. & Macknik, S. L. The impact of microsaccades on vision: towards a unified theory of saccadic function. *Nature Reviews Neuroscience* **14**, 83-96 (2013).
- 4 Schneider, R. M. *et al.* Neurological basis for eye movements of the blind. *PloS one* **8**, e56556 (2013).
- 5 Leigh, R. & Zee, D. S. Eye movements of the blind. *Investigative Ophthalmology & Visual Science* **19**, 328-331 (1980).

Comment 2: All experiments in this paper were performed on rats. The size of the human eyeball is much larger than that of the rat eyeball, so the ultrasonic waves reach the retina over a longer distance. Since the attenuation of ultrasonic waves increases with increasing frequency, the attenuation may not be negligible at high resolution. The generation of heat is also a concern. Consideration is needed regarding the appropriate frequency and spatial resolution that can be expected to be achieved when this device is applied to actual humans.

Response to Comment 2: Thank you for raising this great point to consider the size and shape difference between humans' and rats' eyeballs. To address this, we have conducted both analytical calculations and FEM simulations. These efforts demonstrate that acoustic attenuation within the human eyeball remains relatively low, ensuring that heating effects are not a concern. The fundamental reason behind this somewhat counterintuitive finding lies in the remarkably low acoustic attenuation properties of the vitreous body, which is 70-100 times lower than that of other tissues and occupies the majority of the volume within the human eyeball.

As shown in our updated **Table S2** (Table R1, added parameters for aqueous and iris), acoustic attenuation of aqueous and vitreous is 0.01 – 0.012 dB/cm/MHz and 0.78 – 1.19 dB/cm/MHz in other tissues. Given the average structure of human eyeball, we can do a quick estimation of the attenuation distribution in the human eyeball with minor mathematical simplifications:

$$\text{Attenuation from aqueous and vitreous} = 0.012 \text{ dB/cm/MHz} * 20 \text{ MHz} * 2 \text{ cm} = 0.48 \text{ dB}$$

$$\text{Attenuation from sclera and iris} = 1.0 \text{ dB/cm/MHz} * 20 \text{ MHz} * 0.1 \text{ cm} = 2 \text{ dB}$$

$$\text{Attenuation from lens} = 1.2 \text{ dB/cm/MHz} * 20 \text{ MHz} * 0.4 \text{ cm} = 9.6 \text{ dB}$$

Given that rats possess relatively large lenses with a center thickness of around 6 mm—surpassing that of humans, which is approximately 4 mm—the overall acoustic attenuation in the eyeballs of both rats and humans is comparable. To further reduce attenuation in human eyeballs, we plan to develop a ring-shaped ultrasound array featuring a central hole to circumvent the human lens. This innovative design not only prevents ultrasound waves from passing through the lens but also safeguards the remaining vision of patients by ensuring the optical path remains clear. Our team has devised a specialized algorithm for generating patterns for this ring array⁴ and is currently advancing towards device fabrication.

To more accurately quantify acoustic attenuation and ultrasound-induced heat generation in human eyeballs, we conducted FEM simulations at 20 MHz (1.1 MPa, 15 ms duration), using tissue parameters from **Table S2**. The results, as depicted in Fig. R1a&b, reveal that the acoustic pressure at the retina location stands at about 1.3 MPa in a free field and 1.1 MPa after considering the attenuation in the human eyeball, corresponding to a 1.45 dB attenuation. Furthermore, Fig. R1c demonstrates that the temperature increase in the human eyeball under 20 MHz ultrasound stimulation remains below 1 degree, reinforcing the safety and efficacy of our approach.

A new note discussing the acoustic attenuation in the human eyeball has been added to the supplementary information as the **Supplementary Note 2**.

Supplementary Note 2: Acoustic attenuation in human eyeball.

Given the significant difference between the human eyeball (diameter around 24 mm) and the rat eyeball (diameter around 7 mm), a common concern is that the acoustic attenuation in the human eyeball will be significantly stronger than the attenuation in the rat eyeball. This attenuation and attenuation-related heating effect over the long distance will make ultrasound retina prostheses infeasible for human applications, especially for high-frequency ultrasound.

Here we performed FEM simulations at 20 MHz (1.1 MPa, 15 ms duration) to quantify the acoustic attenuation and ultrasound-induced heat generation in human eyeballs, demonstrating that acoustic attenuation in human eyeball is still relatively low and the heating effect won't be a concern. As shown in Fig. S1a&b, the acoustic pressure at the retina location is around 1.3 MPa in free field and 1.1 MPa with the attenuation in human eyeball, which indicates a 1.45 dB attenuation. Accordingly, Fig. R1c shows the temperature increase in the human eyeball under 20 MHz ultrasound stimulation is still within 1 degree.

The key reason leading to this counterintuitive phenomenon is the significantly low acoustic attenuation of the vitreous (70-100 times lower than other tissues), which fills the main volume of the human eyeball. As shown in our updated Table S2, acoustic attenuation of aqueous and vitreous is 0.01 – 0.012 dB/cm/MHz and 0.78 – 1.19 dB/cm/MHz in other tissues. Given the average structure of human eyeball, we can do a quick estimation of the attenuation distribution in the human eyeball with minor mathematical simplifications:

$$\text{Attenuation from aqueous and vitreous} = 0.012 \text{ dB/cm/MHz} * 20 \text{ MHz} * 2 \text{ cm} = 0.48 \text{ dB}$$

$$\text{Attenuation from sclera and iris} = 1.0 \text{ dB/cm/MHz} * 20 \text{ MHz} * 0.1 \text{ cm} = 2 \text{ dB}$$

$$\text{Attenuation from lens} = 1.2 \text{ dB/cm/MHz} * 20 \text{ MHz} * 0.4 \text{ cm} = 9.6 \text{ dB}$$

Since rat eyeball has a relatively big lens with a center thickness of around 6 mm, which is even thicker than human's lens (~ 4 mm), the overall acoustic attenuations in rats' eyeball and humans' eyeball are similar.

Figure R1 (Supplementary Fig. 1). Acoustic attenuation and heat generation of 20-MHz ultrasound in the human eyeball. **a**, Ultrasound field in free space. **b**, Ultrasound field in the human eyeball with attenuation. **c**, Ultrasound-induced temperature increase distribution in the human eyeball.

Table R1 (Table S2 in the manuscript). Parameters of tissues in the eye for acoustic and thermal simulation.

Tissues	Density (kg/m ³)	Sound speed (m/s)	Heat capacity at constant pressure (J/kg/K)	Thermal conductivity (W/m/K)	Attenuation (dB/cm/MHz)
Water	1000	1500	4178	0.62	0
Cornea	1062	1586	4178	0.58	0.78
Aqueous	1007	1497	3997	0.59	0.01
Vitreous	1005	1532	3999	0.6	0.012
Lens	1076	1647	3000	0.40	1.19
Iris*	1090	1588	3421	0.49	0.62
Sclera	1088	1647	4178	0.58	0.97
Retina	1034	1538	3680	0.57	1.15

*Direct measurement results of Iris are not found. Muscle parameters were used for iris since iris mainly composed of muscle.

Reviewer #2 (Remarks to the Author):

General Comment: The presented study by Lu and colleagues on noninvasive ultrasonic stimulation in the context of vision restoration is quite intriguing. The authors utilize an ultrasound-based approach to stimulate the retinae of rats. This is paired with MEA recordings in the superior colliculus and a behavioral assay of light-stimulated anticipatory licking. In addition, the extended data set includes optical recordings of calcium transients in visual cortex. The results beautifully demonstrate that ultrasonic stimulation of the retina is able to produce signals in the superior colliculus that correlate with the presented stimuli on the piezo-ultrasound device. In addition, the behavioral data suggest that these ultrasound stimuli, when directed at the eye, are ultimately received and interpreted as visual stimuli. The authors have additionally addressed some of the health-concerns that may arise from the use of ultrasonic stimulation, although clearly more would need to be done prior to use in humans.

Response to General Comment:

Thank you very much for recognizing our work and providing insightful and constructive feedback. Safety issues and health concerns are among the most critical factors in assessing the feasibility of Ultrasound Retina Prosthesis (U-RP). We have meticulously addressed your comments on a point-by-point basis. Any changes or additions to the manuscript and the supplementary information are highlighted in blue. We trust that our responses adequately address your concerns. Furthermore, we wholeheartedly concur that comprehensive safety studies involving larger animal models and additional data collection are imperative before proceeding to human trials.

Specific Comments

Comment 1: Ultrasound can permeabilize cells, and pressures and durations similar to those used here disrupted the blood-retinal barrier in one 2020 study using focused ultrasound. Have the authors performed an experiment to exclude the possibility of blood-retinal barrier disruption? Is there evidence to rule out extravasation, for example?

Response to Comment 1: We are grateful for the reviewer's attention to this matter. It is noteworthy that, although the ultrasound pressures and durations employed in our study are comparable to, or even exceed, those used for disrupting the blood-retina barrier (BRB) and blood-brain barrier (BBB), the frequency of the ultrasound waves and the use of microbubble contrast agents stand as two additional critical factors in evaluating the risk of ultrasound-induced barrier disruption. After considering these two factors, we don't expect blood-retina barrier disruption induced by the ultrasonic retina prosthesis, underscoring the specificity and safety of our approach in this context.

Since the discovery of this phenomenon earlier than 1956⁷, great efforts have been devoted to finding the threshold and understanding the physical mechanism behind it and fortunately the mechanism has been confirmed by various independent studies, physical theories, and experiments⁸. The mechanism is that the acoustic cavitation effect dominates the cell permeabilization and barrier opening. Acoustic cavitation effect is stronger when the ultrasound frequency is lower. Also, the injection of microbubbles can further enhance the cavitation effect and significantly lower the ultrasound threshold for cavitation⁹. Here, we listed ultrasound parameters from papers regarding barrier disruption with ultrasound:

Reference*	Barrier	Ultrasound frequency	Ultrasound pressure	MI	Microbubbles exist
https://doi.org/10.1371/journal.pone.0042754	BRB	0.69 MHz	1.1 MPa	1.3	Yes
Theranostics. 2020; 10(7): 2982–2999.	BRB	1.1 MHz	0.56-0.84 MPa	0.53 – 0.8	Yes
Pharmaceutics 2022, 14(3), 494	BRB	1.5 MHz	0.5-0.7 MPa	0.41-0.57	Yes
Theranostics. 2012; 2(12): 1223–1237.	BBB	1.5 MHz	0.5 MPa	0.4	Yes
Ultrasound Med Biol. 1995;21(7):969-79. doi: 10.1016/0301-5629(95)00038-s.	BBB	0.936 MHz	7.7 – 10.4 MPa	7.95 - 10.74	No
https://doi.org/10.1016/S0301-5629(01)00521-X	BBB	2 MHz	3.8 MPa	2.68	No

*We failed to find any work that disrupted blood retina barrier without using microbubbles.

We effectively ruled out the risk of blood-retinal barrier disruption by meticulously evaluating the ultrasound pressures and frequencies utilized in our research. Despite our ultrasound pressures being within the range known to open barriers (1 - 4 MPa), our employed frequencies (3 - 20 MHz) are significantly higher than those typically associated with barrier disruption (usually < 1.5 MHz). Further evidence supporting our conclusion is provided by our mechanical index (MI) values, as illustrated in Extended **Data Figure 6a**. The MI, a measure extensively utilized by the FDA to assess and regulate the effects of acoustic cavitation and related safety concerns, remained well within FDA guidelines¹⁰ (MI < 1.9) for frequencies above 3 MHz, dropping to even lower

than 0.23 at around 20 MHz. (We tested the effect of low-frequency ultrasound with frequency from 1 to 3 MHz in the “Physical mechanism of U-RP” study. Although it is possible that a certain level of BRB disruption happened in these frequency range since their MI is higher than 1.9, this frequency range was only used for mechanism study and will not be used in the practical U-RP.)

In summary, given the frequency-dependent pressures employed in our study devoid of microbubbles, alongside our experimental findings, we confidently anticipate that the ultrasonic retina prosthesis did not cause significant disruption to the blood-retina barrier.

To emphasize this point, in the **Discussion** of the revised manuscript, we have added: While the ultrasound pressures used in this study were also reported to permeabilize cells and open blood-retina barrier by other works, it should be noted that the barrier opening phenomena require the injection of microbubble agents, which were not used in this study.

Comment 2: Microglial also express piezo channels that control their activity. The immunolabeling for microglial cells experiment lacks a positive control. How does the reader know that this antibody worked? One would expect to see microglial in the intact rodent retina plexiform layers, so one could demonstrate that there are microglial cells present and then ask whether numbers or distribution changed, in addition to whether or not they became activated.

Response to Comment 2: We thank the reviewer for the professional comments. Following your instructions, we performed immunostaining again and added positive control group. In the positive control group, the 20-MHz ultrasound were applied for three hours with a pressure 100% higher than the safe intensity (3.5 MPa vs 1.7 MPa, 10-ms pulse duration, and a repetition frequency of 5 Hz), in order to elicit an inflammatory response in the retina. Following the stimulation, rats were euthanized on day 3, and immunostaining was performed. The negative control group consisted of normal rats that did not receive ultrasonic stimulation. Rats in the stimulation group were subjected to a two-week ultrasonic stimulation at a safe intensity (as described in the Method). We analyzed the expression of Glial Fibrillary Acidic Protein (GFAP) and Cluster of Differentiation 68 (CD68) in central retinal sections of these different groups.

In the negative control and ultrasound stimulation groups, GFAP expression was detectable in retinal astrocytes and a few Müller cells. In contrast, for retinas in the positive control group, there was a significant increase in the number of GFAP-positive Müller cells, with distribution also observed in the Inner Nuclear Layer (INL). CD68-positive microglia cell bodies were virtually absent in the retinas of the wild-type group and rats subjected to safety ultrasound stimulation. However, in the positive control group, CD68-positive microglia/macrophages were detected in both the Inner Nuclear Layer and the Retinal Ganglion Cell Layer.

Immunostaining results have confirmed that our antibody was indeed effective, and that ultrasonic retina prosthesis is safe, as it did not trigger a significant immune response. We have updated the **Extended Data Fig. 4 and Supplementary Fig. 14:**

Figure R2 (Supplementary Fig. 14). Immunostaining analysis of the ex vivo retina after the ultrasound stimulation with positive control group. An inflammatory response in the retina was intentionally induced in the positive control group.

Figure R3 (Extended Data Fig. 4). U-RP is safe in long-term use.

Comment 3: Figure 5B. it appears that this animal responded to light with anticipatory licks on day 7. However, this animal is also described as being “blind” in the figure legend. Please clarify whether this animal is blind and if so, why it has anticipatory licking in response to light.

Response to Comment 3: Thank you so much for mentioning this point. We sincerely apologize for the misleading figure legend and confusions it raised. Results in Figure 5b are from a blind rat and they only had ultrasound stimulation. We have corrected the **figure legend of Fig. 5b** by changing “light stimulation” to “ultrasound stimulation” in the revised manuscript:

“**b**, Licking responses during representative sessions of a blind rat. Day 1: start of training with ultrasound stimulation. Day 7: end of training with ultrasound stimulation. Day 8: test with ultrasound stimulation. (Ultrasound parameters in both training and testing sessions: 4.5 MHz, 20 ms, 2.9 MPa).”

Just to further clarify our experiment setup here:

1. The behavior experiment flow of blind rats was using ultrasound stimulation through Day 1 to Day 8. Since all blind rats have been confirmed to be totally blind (Fig. 1e), light stimuli were not used on blind rats.
2. The behavior experiment flow of normal rats: On Days 1-7, training with light stimulation. On Day 8, test with ultrasound stimulation.

Comment 4: Figure 5C. Please clarify in the figure legend the day corresponding to the data that is shown?

Response to Comment 4: Figure 5C shows the data from testing sessions, i.e., Day 8. We have added in the figure legend of **Fig. 5c**:

“c, Lick response heatmaps recorded in Day 8.”

Comment 5: Figure 5D. Earlier in the figure, different sessions across several days are shown. To which sessions or days do these data refer?

Response to Comment 5: We apologize for the confusion due to the misleading label. The x-axis label of Figure 5d has been corrected from “session” to “Day”. To further clarify, the “session” was used to indicate the ultrasound stimulation period which typically lasts for 30 minutes on each animal per day. Day 1 to Day 7 hold training sessions and Day 8 holds the testing session.

Comment 6: It is not always clear how many animals were studied for each condition for each set of studies. Please provide this information for each set of experiments. Figure 1, for examples, provides the number of neurons, but not the number of animals. One would like to know that data are reproducible across animals.

Response to Comment 6: Thank you for your constructive comments. We have clarified this information in the **Method – Animals** of the revised manuscript:

Three blind rats and three healthy rats were used in the initial validation of both light and ultrasound stimulation. Eight blind rats were used to investigate the frequency-dependent acoustic threshold and spatial-temporal resolution investigation, and pattern generation. Eight blind and eight healthy rats were used in the water-licking behavior tests. Two blind rats and two healthy rats were used in the in vivo fiber photometry experiment. Three healthy rats were used in the safety examination study. The unstimulated eye was used in the negative control group. One healthy rat was used in the positive control group.

Comment 7: Supplemental Figure 13 shows changes in gene expression for several proteins. Could these changes be indicative of pathology or a response to a noxious stimulus? What is the significance of the changes in Itgb3, for example?

Response to Comment 7: Thank you for pointing out this important and interesting question. From the bulk RNA seq test, we found that in our ultrasound stimulation model, mRNAs that are upregulated are mainly related to responses to mechanical stimulation. In Supplemental Figure 13, we showed the upregulation of multiple genes including piezo1, Bag3, Cnn2, Nos3, Itgb3, and tgfb1. All these genes are involved in the responses to the mechanical stimulation pathway. For example, Bag3 is known to be involved in apoptosis, development, cytoskeleton organization, and autophagy, thereby mediating cell adaptive responses to stressful stimuli¹¹; Cnn2 is sensitively regulated by mechanical tension, influenced by the mechanics within the cytoskeleton¹²⁻¹⁴; Nos3 can be activated by mechanical activity¹⁵; tgfb1, mechanical activity can induce the expression of tgfb1¹⁶. For Piezo1, it is a mechanically gated ion channel known for converting mechanical stimuli into electrical signals in mammals¹⁷. Although no direct indication of the upregulation of the beta 3 integrin (Itgb3) is due to mechanical stimulation, researchers have shown the upregulation of Itgb3 can be a downstream result of the regulation of the Piezo1 receptor. Previous studies suggest that Piezo1 mechanoenzyme regulates integrin-dependent chemotactic migration in human T cells¹⁸.

In addition, we did literature research to see whether the upregulation of genes related to responses to mechanical stimulation observed in this study is a common response to other types of noxious stimulation, or a unique response to ultrasound stimulation. We reviewed a vast amount of literature and compared our results with studies examining gene regulation after various types of extrinsic stimulation to the retina. These studies include measuring the gene regulation after the application of chemical toxin to

the retina, and showing that mRNAs were mainly enriched in signaling pathways related to immune inflammation¹⁹; in the ischemia–reperfusion injury model, mRNAs were mainly enriched in signaling pathways related to the apoptotic signaling pathway, regulation of neuron death, and Oxidative stress²⁰; in the high-powered laser exposure model, mRNAs were mainly enriched in signaling pathways related to complement activation, glial/macrophage activation, and inflammatory pathways²¹. Unlike ultrasound stimulation, none of the other stimulation models showed a regulation of mechanical response-related genes. By comparing our results with these studies, we can conclude that the up-regulation of the mechanical-response related genes observed in our ultrasound stimulation model is unique and distinctly different from the responses obtained with other stimulations.

Furthermore, to rule out the possibility of pathology or noxious effect, we performed comprehensive safety examinations (OCT, ERG, Immunostaining), and demonstrated that no damage was induced by ultrasound retina stimulation.

Minor Comments

Minor Comment 1: The similarity of the ERG a and b waves in controls versus rats exposed to multiple episodes of ultrasound stimulation is encouraging. However, these waves report activity of photoreceptors and bipolar cells and not retinal ganglion cells. If the ultrasound waves are impacting neurons or cells other than photoreceptors and bipolar, that could not be readily detected by analyzing the a and b waves.

Response to Minor Comment 1: Thank you for your professional feedback. We concur that while the ERG results substantiate the functional safety of ultrasound stimulation on photoreceptors and bipolar cells, they do not conclusively affirm the safety for other cell types, including retinal ganglion cells (RGCs). The health of RGCs has been partially evidenced through additional findings from histology, OCT, and Fundus imaging, as well as functionally, through their capability to receive visual signals from the brain.

Furthermore, we have reason to believe that RGC function remains intact following ultrasound stimulation. This is supported by our observations of repetitive and stable neuronal responses in the brain after repeated ultrasound stimulations during our investigations into the temporal resolution of U-RP. Given that RGCs are responsible for transmitting signals from the lower layers of the retina to the optic nerve and brain, any impairment in RGC function would likely result in compromised neuronal signals in the brain. In light of this, we have revised the statement in the legend of **Extended Data Fig. 4c** and in the **Safety examination of U-RP** section of our manuscript to more accurately reflect our findings and conclusions:

Secondly, we measured electroretinography (ERG) responses to full-field light stimuli to evaluate any potential functional damage in photoreceptors and bipolar cells caused by ultrasound stimulation (Extended Data Fig. 4c). Secondly, we measured electroretinography (ERG) responses to full-field light stimuli to evaluate any potential functional damage in photoreceptors and bipolar cells caused by ultrasound stimulation (Extended Data Fig. 4c). Neither the amplitudes nor time delays of a- and b-waves showed any statistically significant changes (Extended Data Fig. 4d, n = 8, s.d.), demonstrating intact function of photoreceptors and bipolar cells. Since repetitive and stable neuron responses were observed from the brain during and after ultrasound stimulation, the function of RGCs was inferred to be intact as well.

Extended Data Fig. 4. U-RP is safe in long-term use. c, In vivo ERG measurement of photoreceptors and bipolar cells function before and after the ultrasound stimulation. In the control group, the ultrasound stimulation was focused between the eye and ear of the rat.

Minor Comment 2: Why were all the rats that were used male?

Response to Minor Comment 2: Since no significant differences were reported in neuronal, structural, or functional aspects between male and female rats within both the retina and visual circuits, we do not anticipate disparate outcomes between sexes in retina stimulation experiments. Therefore, selecting a single sex for this experiment should not compromise the scientific integrity or gender equity of our research. Considering these factors, we opted to use only one sex—male rats—primarily for convenience in the experimental setup. The choice of male rats was influenced by the potential impact of the estrous cycle on the behavior of female rats in water-licking experiments. Additionally, the comparatively larger body size of male rats facilitates more effective restraint during head and body fixation procedures.

Minor Comment 3: Figure 1f – correct spelling of healthy

Response to Minor Comment 3: Thank you for pointing this out. We have corrected it in the revised manuscript.

Minor Comment 4: Figure 5E is difficult to visually interpret. The lines connecting each condition detracts from the actual data. Also, what do the grey boxes in between baseline and anticipatory licks represent?

Response to Minor Comment 4: We appreciate your constructive feedback. Following your suggestions, we have updated Fig. 5e in the revised manuscript. In the new version, lines are directly connected to each data point. The previous grey boxes baseline and anticipatory licks were lines. We hope the new version of Fig. 5e is clearer. The new version of Fig. 5 is below:

Fig. 5. Behavior response induced by the ultrasound retina stimulation. **a**, Experiment setup of water-licking behavior tests. Light or ultrasound stimulation of one eye cues water drops. Animals responded by licking before (anticipatory licks) or after the appearance of water. **b**, Licking responses during representative sessions of a blind rat. Day 1: start of training with ultrasound stimulation. Day 7: end of training with ultrasound stimulation. Day 8: test with ultrasound stimulation. (Ultrasound parameters in both training and testing sessions: 4.5 MHz, 20 ms, 2.9 MPa). **c**, Lick response heatmaps recorded in Day 8. Rows: responses of different animals. Columns: responses in one-second time bins. Top: ultrasound stimulation group. Bottom: control group with sham stimulation behind the eye. Left: healthy rats. Right: blind rats. **d**, Average anticipatory lick rates as along the number of training sessions. Top: the results from blind rats that were stimulated by ultrasound in both training and testing sessions. Bot: the

results from healthy rats that were trained with light stimulation and tested with ultrasound stimulation. e, Overall analysis of Mean lick rates in different stages on Day 8. Error bars: s.e.m, $n = 8$; center lines: median values.

Reference

- 1 Land, M. & Tatler, B. *Looking and acting: Vision and eye movements in natural behaviour*. (Oxford University Press, 2009).
- 2 Hessels, R. S., Niehorster, D. C., Nyström, M., Andersson, R. & Hooge, I. T. Is the eye-movement field confused about fixations and saccades? A survey among 124 researchers. *Royal Society open science* **5**, 180502 (2018).
- 3 Martinez-Conde, S., Otero-Millan, J. & Macknik, S. L. The impact of microsaccades on vision: towards a unified theory of saccadic function. *Nature Reviews Neuroscience* **14**, 83-96 (2013).
- 4 Lu, J.-y., Lu, G., Thomas, B. B., Humayun, M. S. & Zhou, Q. Ultrasound Concave 2D Ring Array for Retinal Stimulation. *IEEE Transactions on Ultrasonics, Ferroelectrics, and Frequency Control* (2023).
- 5 Schneider, R. M. *et al.* Neurological basis for eye movements of the blind. *PloS one* **8**, e56556 (2013).
- 6 Leigh, R. & Zee, D. S. Eye movements of the blind. *Investigative Ophthalmology & Visual Science* **19**, 328-331 (1980).
- 7 Bakay, L., Hueter, T., Ballantine, H. & Sosa, D. Ultrasonically produced changes in the blood-brain barrier. *AMA Archives of Neurology & Psychiatry* **76**, 457-467 (1956).
- 8 Sheikov, N., McDannold, N., Vykhodtseva, N., Jolesz, F. & Hynynen, K. Cellular mechanisms of the blood-brain barrier opening induced by ultrasound in presence of microbubbles. *Ultrasound in medicine & biology* **30**, 979-989 (2004).
- 9 Choi, J. J. *et al.* Microbubble-size dependence of focused ultrasound-induced blood-brain barrier opening in mice in vivo. *IEEE Transactions on Biomedical Engineering* **57**, 145-154 (2009).
- 10 Şen, T., Tüfekçioğlu, O. & Koza, Y. Mechanical index. *Anatolian journal of cardiology* **15**, 334 (2015).
- 11 Rosati, A., Graziano, V., De Laurenzi, V., Pascale, M. & Turco, M. C. BAG3: a multifaceted protein that regulates major cell pathways. *Cell death & disease* **2**, e141-e141 (2011).
- 12 Hossain, M. M., Crish, J. F., Eckert, R. L., Lin, J. J.-C. & Jin, J.-P. h2-Calponin is regulated by mechanical tension and modifies the function of actin cytoskeleton. *Journal of Biological Chemistry* **280**, 42442-42453 (2005).
- 13 Hossain, M. M., Smith, P. G., Wu, K. & Jin, J.-P. Cytoskeletal tension regulates both expression and degradation of h2-calponin in lung alveolar cells. *Biochemistry* **45**, 15670-15683 (2006).
- 14 Jiang, W.-r. *et al.* Mechanoregulation of h2-calponin gene expression and the role of Notch signaling. *Journal of Biological Chemistry* **289**, 1617-1628 (2014).
- 15 Kaye, D. M., Wiviott, S. D. & Kelly, R. A. Activation of nitric oxide synthase (NOS3) by mechanical activity alters contractile activity in a Ca²⁺-independent manner in cardiac myocytes: role of troponin I phosphorylation. *Biochemical and biophysical research communications* **256**, 398-403 (1999).
- 16 Manokawinchoke, J. *et al.* Mechanical force-induced TGFB1 increases expression of SOST/POSTN by hPDL cells. *Journal of Dental Research* **94**, 983-989 (2015).
- 17 Atcha, H. *et al.* Mechanically activated ion channel Piezo1 modulates macrophage polarization and stiffness sensing. *Nature communications* **12**, 3256 (2021).
- 18 Liu, C. S. C. *et al.* Piezo1 mechanosensing regulates integrin-dependent chemotactic migration in human T cells. *Elife* **12**, RP91903 (2024).
- 19 Chen, S. *et al.* RNA-seq analysis reveals differentially expressed inflammatory chemokines in a rat retinal degeneration model induced by sodium iodate. *Journal of International Medical Research* **50**, 03000605221119376 (2022).
- 20 Li, Y. *et al.* Single-cell RNA sequencing reveals a landscape and targeted treatment of ferroptosis in retinal ischemia/reperfusion injury. *Journal of Neuroinflammation* **19**, 261 (2022).
- 21 Ibbett, P. *et al.* A lasered mouse model of retinal degeneration displays progressive outer retinal pathology providing insights into early geographic atrophy. *Scientific Reports* **9**, 7475 (2019).

REVIEWERS' COMMENTS

Reviewer #1 (Remarks to the Author):

The authors have adequately addressed my earlier remarks in this revision. I have no further comments.